# A mouse model of human mitofusin-2-related lipodystrophy exhibits adipose-specific mitochondrial stress and reduced leptin secretion

Jake P Mann[1], Xiaowen Duan[1], Satish Patel[1], Luis Carlos Tábara[2], Fabio Scurria[1], Anna Alvarez-Guaita[1], Afreen Haider[1], Ineke Luijten[3], Matthew Page[4], Margherita Protasoni[2], Koini Lim[1], Sam Virtue[1], Stephen O'Rahilly[1], Martin Armstrong[5], Julien Prudent[2], Robert K Semple[3,6]*†, David B Savage[1]*†

[1]Wellcome Trust-MRC Institute of Metabolic Science, University of Cambridge, Cambridge, United Kingdom; [2]Medical Research Council Mitochondrial Biology Unit, University of Cambridge, Cambridge, United Kingdom; [3]Centre for Cardiovascular Science, University of Edinburgh, Edinburgh, United Kingdom; [4]New Medicines, UCB Pharma, Slough, United Kingdom; [5]UCB Pharma, Chemin du Foriest, Braine l'Alleud, Belgium; [6]MRC Human Genetics Unit, University of Edinburgh, Edinburgh, United Kingdom

*For correspondence:
rsemple@exseed.ed.ac.uk (RKS);
dbs23@cam.ac.uk (DBS)

†These authors contributed equally to this work

Competing interest: The authors declare that no competing interests exist.

**Abstract** Mitochondrial dysfunction has been reported in obesity and insulin resistance, but primary genetic mitochondrial dysfunction is generally not associated with these, arguing against a straightforward causal relationship. A rare exception, recently identified in humans, is a syndrome of lower body adipose loss, leptin-deficient severe upper body adipose overgrowth, and insulin resistance caused by the p.Arg707Trp mutation in *MFN2*, encoding mitofusin 2. How the resulting selective form of mitochondrial dysfunction leads to tissue- and adipose depot-specific growth abnormalities and systemic biochemical perturbation is unknown. To address this, $Mfn2^{R707W/R707W}$ knock-in mice were generated and phenotyped on chow and high fat diets. Electron microscopy revealed adipose-specific mitochondrial morphological abnormalities. Oxidative phosphorylation measured in isolated mitochondria was unperturbed, but the cellular integrated stress response was activated in adipose tissue. Fat mass and distribution, body weight, and systemic glucose and lipid metabolism were unchanged, however serum leptin and adiponectin concentrations, and their secretion from adipose explants were reduced. Pharmacological induction of the integrated stress response in wild-type adipocytes also reduced secretion of leptin and adiponectin, suggesting an explanation for the in vivo findings. These data suggest that the p.Arg707Trp MFN2 mutation selectively perturbs mitochondrial morphology and activates the integrated stress response in adipose tissue. In mice, this does not disrupt most adipocyte functions or systemic metabolism, whereas in humans it is associated with pathological adipose remodelling and metabolic disease. In both species, disproportionate effects on leptin secretion may relate to cell autonomous induction of the integrated stress response.

## Editor's evaluation

This work describes a mouse model of a human mitofusin 2- related lipodystrophy, generated by knockin of Mfn2 R707W, and reports adipocyte-specific effects involving activation of a cellular integrated stress response and consequently reduced secretion of leptin and adiponectin. The

phenotypic characterization is thorough, and the data are convincing. The work provides important information to link mitochondrial perturbation selectively with altered adipose function, and to show how these effects contribute to metabolic disease.

## Introduction

Mitochondrial dysfunction has been implicated in the pathogenesis of a wide range of congenital and acquired conditions (*Gorman et al., 2016*; *Thompson et al., 2020*; *Craven et al., 2017*). However, despite being central to cellular energy homeostasis, there has been little mechanistic evidence of a causal role for deranged mitochondrial function in human adiposity. Instead, most patients with inherited mitochondrial disorders have a neurological phenotype, although multisystem involvement is common (*Gorman et al., 2016*; *Suomalainen and Battersby, 2018*). This is true for disorders caused by many different mutations affecting mitochondrial proteins, whether encoded by the mitochondrial or nuclear genome. The mechanisms underlying such tissue-selective disease manifestations even in the face of constitutional mutations are unclear, but differing tissue requirements for oxidative phosphorylation, and interactions between the nuclear and mitochondrial genome may play a role (*Pacheu-Grau et al., 2018*; *Alston et al., 2017*; *Schon and Manfredi, 2003*).

We (*Rocha et al., 2017*) and others (*Sawyer et al., 2015*; *Capel et al., 2018*; *Carr et al., 2015*; *Masingue et al., 2017*; *Piscosquito et al., 2015*) recently identified biallelic R707W mutations in the nuclear *MFN2* gene in patients with a remarkable adipose phenotype characterised by extreme upper body adiposity (lipomatosis) and lower body lipodystrophy. Affected patients also showed non-alcoholic fatty liver disease, dyslipidaemia and insulin resistant type 2 diabetes, likely secondary to the changes in adipose tissue. *MFN2* encodes mitofusin 2, a mitochondrial outer membrane protein that plays a key role in mitochiondrial fusion and tethering to other organelles (*Giacomello et al., 2020*). Like patients with heterozygous complete loss-of-function mutations in *MFN2*, patients with the R707W mutation also often exhibit axonal peripheral neuropathy known as Charcot-Marie Tooth type 2 A (CMT2A) (*Chung et al., 2006*; *Verhoeven et al., 2006*; *Neusch et al., 2007*), but the adipose and metabolic phenotype has uniquely been associated with the R707W allele to date. Evidence suggesting that the MFN2 R707W mutation does disrupt mitochondrial function in humans includes elevated serum lactate in affected patients, abnormal mitochondrial morphology seen on transmission electron microscopy of affected adipose tissue (*Rocha et al., 2017*), and strong transcriptomic signatures of mitochondrial dysfunction and activation of the integrated stress response (ISR) in the same tissue.

Despite normal or raised whole body adipose mass, and the relatively normal histological appearance of lipomatous upper body fat, plasma leptin concentrations were very low in the patients reported by Rocha et al (*Rocha et al., 2017*). This observation was supported by Sawyer et al. who reported one patient with undetectable serum leptin (<0.6 ng/mL) (*Sawyer et al., 2015*) and Capel et al. who described five patients with serum leptin concentrations <1.6 ng/mL despite body mass indices (BMIs) within the normal range (*Capel et al., 2018*). Leptin is a critical endocrine signal of adipose stores and yet what determines the rate of adipocyte leptin secretion remains poorly understood (*Szkudelski, 2007*). This surprising observation thus offered a rare opportunity to address this important issue. Circulating leptin concentration correlates with fat mass, and is usually higher in women than men, with some adipose depots reported to release more than others (*Montague et al., 1997*). Leptin secretion is increased by insulin stimulation (*Cammisotto et al., 2005*) but this effect is modest in the context of serum leptin concentrations. Adipose depots that secrete higher leptin have increased *LEP* mRNA and, at least in vitro, *LEP* mRNA increases in response to insulin stimulation (*Alvarez-Guaita et al., 2021*).

Most neuropathy-associated *MFN2* mutations are located within the protein's GTPase domain (*Kijima et al., 2005*; *Züchner et al., 2004*; *Feely et al., 2011*; *Stuppia et al., 2015*), but to date, all patients with *MFN2*-associated multiple symmetric lipomatosis (MSL) have had at least one R707W allele. Most cases have been homozygous for the R707W mutation, with others compound heterozygous for R707W and a second, functionally null mutation (*Rocha et al., 2017*; *Sawyer et al., 2015*; *Capel et al., 2018*; *Carr et al., 2015*; *Masingue et al., 2017*; *Piscosquito et al., 2015*). Arginine 707 lies in the highly conserved heptad-repeat (HR)2 region of MFN2 but consensus on the precise orientation and/or function of this domain has not yet been established (*Mattie et al., 2018*). The

dominant model holds that the HR2 domains of Mitofusin 1 (MFN1) and/or MFN2 face the cytosol and interact in trans, forming mitofusin homodimers or heterodimers required for mitochondrial fusion and tethering to other organelles (**Koshiba et al., 2004**; **Franco et al., 2016**). The R707W mutation may disrupt this binding and subsequent MFN oligomerisation and mitochondrial fusion/tethering.

Global knock-out of the core mitochondrial fusion-fission machinery proteins *Mfn1*, *Mfn2*, *Opa1*, and *Drp1* in mice confers embryonic lethality in all cases (**Davies et al., 2007**; **Ishihara et al., 2009**; **Chen et al., 2003**). Two homozygous loss-of-function *Mfn2* knock-ins - H361Y and R94W - have also been reported. H361Y was also embryonically lethal due to complete loss of detectable Mfn2 protein (**Misko et al., 2010**), while homozygous R94W mice died at post-natal day 1 (**Strickland et al., 2014**). Mfn2$^{R94W}$ is expressed but is GTPase defective, increasing mitochondrial fragmentation and preventing formation of Mfn2 homodimers (**Li et al., 2019**).

Multiple tissue-specific knock-outs of *Mfn2* (and/or *Mfn1*) have been studied, including three in adipose tissue (**Chen et al., 2007**; **Boutant et al., 2017**; **Mahdaviani et al., 2017**; **Mancini et al., 2019**). Adipose-specific *Adipoq*::Cre *Mfn2* knock-out increased fat mass, attributed to reduced energy expenditure, whether it occurred in embryonic life (**Boutant et al., 2017**) or was induced by tamoxifen in adult mice (**Mancini et al., 2019**), with ultrastructural evidence of more rounded mitochondria with fewer lipid droplet contacts (**Boutant et al., 2017**). Both Adipoq::Cre driven (adipose-specific) and *Ucp1*::Cre driven (brown adipose-specific) *Mfn2* knock-out also caused cold intolerance with 'whitened' brown adipose tissue (**Mahdaviani et al., 2017**). Plasma leptin concentrations were not reported in Adipoq-Cre or *Ucp1*::Cre *Mfn2* knock-outs, but leptin concentration was higher and adiponectin concentration lower in the tamoxifen-inducible adult Adipoq::Cre *Mfn2* knockout model (**Mancini et al., 2019**).

These studies suggest that *Mfn2* has a non redundant role in adipose tissues, but findings to date are not readily reconcilable with the phenotype observed in humans with the MFN2$^{R707W}$ mutation. We thus generated and characterised mice homozygous for Mfn2$^{R707W}$ to determine the extent of tissue-specific manifestations of mitochondrial dysfunction and to interrogate the effect of this mutant on adipose leptin secretion.

## Results

### Generation of Mfn2$^{R707W/R707W}$ mice

*Mfn2*$^{R707W/R707W}$ mice were generated by CRISPR-Cas9 genomic engineering using an ssODN (single-stranded oligo donor) template recoding Arginine to Tryptophan in codon 707 (**Figure 1A**). A single round of targeting yielded one founder (F0) mouse (**Figure 1B**) which was used to expand a colony on a C57/BL6J background. An additional silent mutation introducing an EcoRV restriction site was introduced to facilitate genotyping (**Figure 1C and D**). We first compared expression of Mfn2 and its paralogue, Mfn1, in knock-in (KI) mice and wild-type (WT) littermates in order to determine if the R707W change perturbed expression of the mutant Mfn2 protein and/or resulted in a compensatory change in Mfn1. We observed no consistent differences in Mfn1 or Mfn2 expression, relative to WT, in inguinal white adipose tissue (WAT), liver, heart, or skeletal muscle in both chow and high-fat diet (HFD) fed mice (**Figure 1E**, **Figure 1—figure supplements 1–4**). However, in brown adipose tissue (BAT) expression of Mfn1 was lower in KI than in WT mice fed with chow diet for 6 months (**Figure 1—figure supplements 1D and 2D**). Similarly, expression of Mfn1 was lower in KI than WT mice after only 4 weeks of chow diet (**Figure 1—figure supplements 1G and 2G**). In HFD-fed mice, both Mfn1 and Mfn2 expression was lower in BAT in KI compared to WT mice (**Figure 1E**, **Figure 1—figure supplements 3D and 4D**). In epididymal WAT, expression of Mfn1 was also lower in KI than in WT mice fed with chow diet (**Figure 1—figure supplements 1F and 2F**), but no differences were observed in HFD-fed mice (**Figure 1—figure supplements 3F and 4F**). We interpret these data as suggesting that the R707W missense mutation does not directly reduce expression of Mfn2, nor does it result in a compensatory change in Mfn1 in most tissues. However, in BAT the data suggests that the cellular perturbation induced in brown adipocytes is associated with a very modest reduction in expression of Mfn1 and 2.

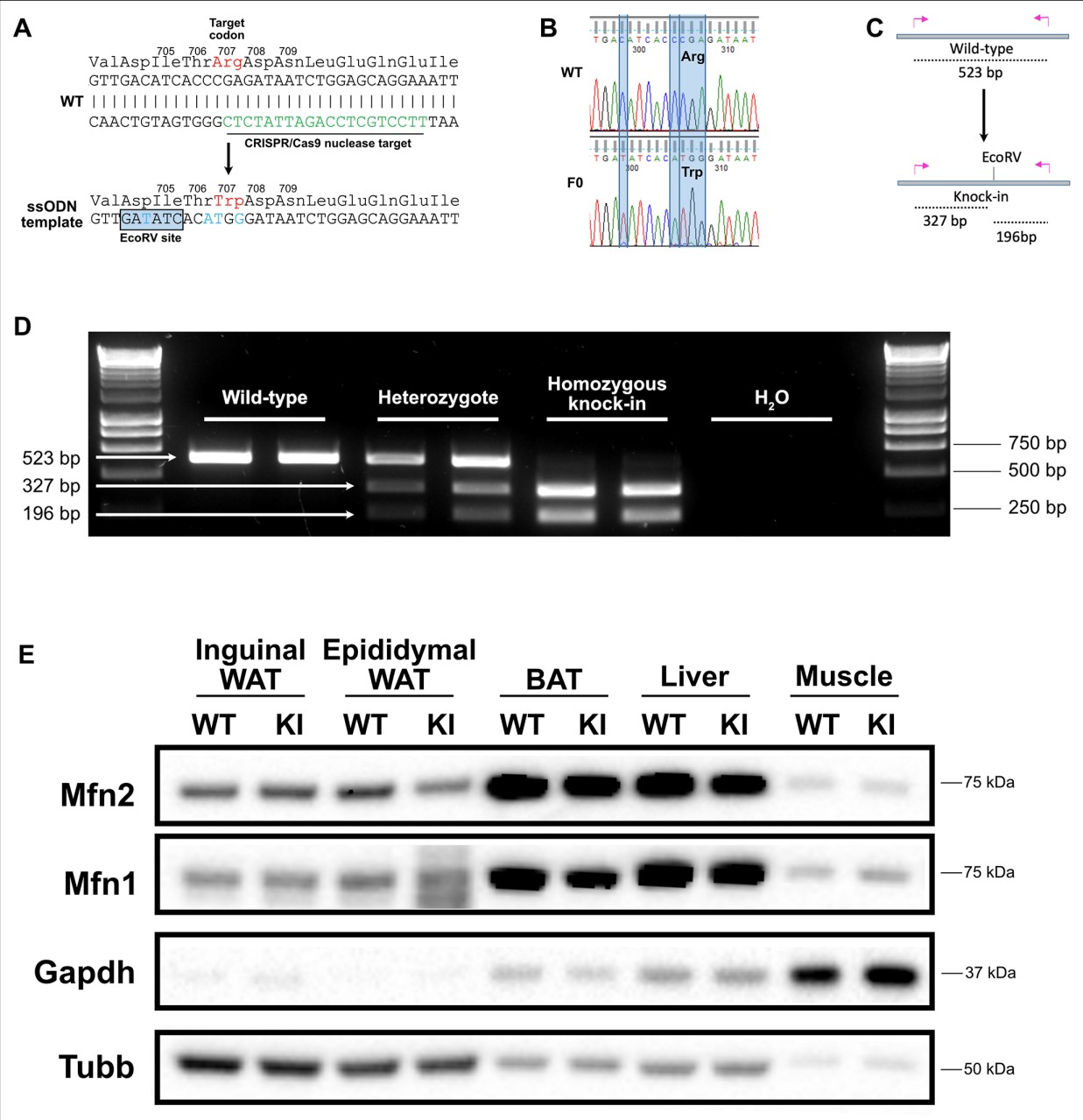

**Figure 1.** Generation of a Mfn2[R707W] knock-in mouse. (**A**) Wild-type (WT) nucleotide and amino acid sequence around the Arg 707 codon. The CRISPR/Cas9 nuclease target is indicated in *green*. Below is part of the ssODN template with mutated nucleotides in *blue*, including the upstream silent mutation (at codon 704–705) to generate an EcoRV restriction site. (**B**) Sanger sequencing confirmation of the knock-in (KI) with restriction site in a founder (**F0**). (**C**) Illustration of the genotyping strategy: mutant alleles will digest into 327 bp and 196 bp fragments in response to EcoRV digestion. (**D**) Ear biopsies were digested using chelix and *Mfn2* amplified by PCR, then digested using EcoRV. Representative SYBR Safe DNA gel demonstrating genotyping for two WT, heterozygous, and homozygous KI mice. Image is representative of other genotyping gels. (**E**) Western blot from inguinal and epididymal white adipose tissue (WAT), brown adipose tissue (BAT), liver and skeletal muscle for expression of Mfn1 and Mfn2. Tissues are from WT and homozygous Mfn2[R707W] KI mice fed a 45% kcal high fat diet (HFD) for 6 months. Due to variability across tissues, both *Gapdh* and Beta-tubulin (*Tubb*) are given as loading controls. The image is representative of at least three biological replicates.

The online version of this article includes the following source data and figure supplement(s) for figure 1:

**Source data 1.** Raw and annotated immunoblots from *Figure 1E*.

**Figure supplement 1.** Expression of mitofusins on chow diet.

**Figure supplement 1—source data 1.** Raw and annotated immunoblots from *Figure 1—figure supplement 1*.

*Figure 1 continued on next page*

*Figure 1 continued*

**Figure supplement 2.** Quantification of western blots for mitofusins in tissues of chow-diet-fed mice.

**Figure supplement 3.** Expression of mitofusins on high-fat diet.

**Figure supplement 3—source data 1.** Raw and annotated immunoblots from *Figure 1—figure supplement 3*.

**Figure supplement 4.** Quantification of western blots for mitofusins in tissues of high-fat diet-fed mice.

## Mfn2$^{R707W/R707W}$ mice show adipose-selective alterations of mitochondrial structure and function

Next, we used transmission electron microscopy (TEM) to examine mitochondrial ultrastructure (*Figure 2A*). In BAT, mitochondria from KI mice had a tendency to exhibit a decreased mitochondrial perimeter compared to WT mice (*Figure 2B*), but significantly reduced size, assessed by the mitochondrial cross-sectional length/ width aspect ratio analysis (*Figure 2C*), indicating that Mfn2$^{R707W}$ leads to mitochondrial fragmentation in BAT. Double membrane-bound structures representing autophagosomes consistent with mitophagy were observed in lipomatous adipose tissue from human patients with the MFN2$^{R707W}$ mutation (*Rocha et al., 2017*), but these were not identified in the murine tissues examined. Mitofusins may mediate contact between mitochondria and lipid droplets (*Boutant et al., 2017*), and the extent of these contacts was reduced in BAT from KI animals (*Figure 2D*). In addition, mitochondrial cristae were disrupted in KI compared to WT animals (*Figure 2E*). In both inguinal (*Figure 2—figure supplement 1A–C*) and epididymal WAT (*Figure 2—figure supplement 1D–F*), similar mitochondrial fragmentation and cristae defects were observed. In contrast, no change in mitochondrial morphology, cristae number, or cristae structure was seen in the heart, skeletal muscle, or liver of KI mice (*Figure 2—figure supplement 2*).

Perturbed mitochondrial dynamics have been associated with decreased mitochondrial DNA (mtDNA) content, as replication of mtDNA relies on balanced fusion and fission (*Silva Ramos et al., 2019*). In Mfn2$^{R707W/R707W}$ mice, mtDNA was reduced in BAT in both diet conditions, but not in any other tissue analysed (WAT, heart, skeletal muscle, or liver) (*Figure 2F–G*). Immunoblotting of electron transport chain components from chow fed animals showed no changes in liver or heart (*Figure 2—figure supplement 3A–B* and *Figure 2—figure supplement 4A–B*). However, mtCo1 (complex IV) and Ndufb8 (complex I) were reduced in KI mice in brown and white adipose tissue (*Figure 2—figure supplement 3C–E* and *Figure 2—figure supplement 4C–E*). In addition, Uqcrc2 (complex III) was also lower in inguinal WAT.

To determine if these changes altered mitochondrial oxidative phosphorylation, we assessed oxidative capacity in freshly isolated mitochondria from BAT and liver by high-resolution respirometry using Oroboros Oxygraphy. No significant differences were detected between WT and KI mitochondria (*Figure 2—figure supplement 5A–B*). We further assessed mitochondrial function in BAT in vivo by challenging mice with noradrenaline in cold (10 degrees) or thermoneutral (30 degrees) conditions to determine maximum thermogenic capacity. Again, despite a trend towards reduced thermogenic capacity in KI animals, the difference was not significant, and both groups manifested the expected increase in energy expenditure at 10 °C (*Figure 2—figure supplement 5C–G*).

## Body composition and metabolic phenotype of Mfn2$^{R707W/R707W}$ mice

We next assessed whether *Mfn2$^{R707W/R707W}$* mice phenocopy the severely abnormal adipose topography and metabolic abnormalities of patients harbouring the same mutations. Male mice fed with either chow or HFD for up to 6 months were assessed. Whole body mass and composition, and masses of individual adipose depots and other organs were similar in KI and WT mice throughout the study period (*Table 1*, *Figure 3A–C* and *Figure 3—figure supplement 1A–D*). Moreover, no difference in hepatic steatosis nor lipid droplet size was detected histologically in BAT or WAT (*Figure 3D–E* and *Figure 3—figure supplement 1F–G*). In keeping with the normal body composition, fasting serum glucose, insulin, triglycerides, cholesterol, lactate, and liver transaminase concentrations showed no difference between WT and KI mice (*Figure 3F–I* and *Figure 3—figure supplement 1E, H*). There was also no differences in hepatic expression of genes related to lipid metabolism or steatohepatitis (*Figure 3—figure supplement 1*). Dynamic testing of glucose and insulin tolerance was also similar between genotypes (*Figure 3G–K*).

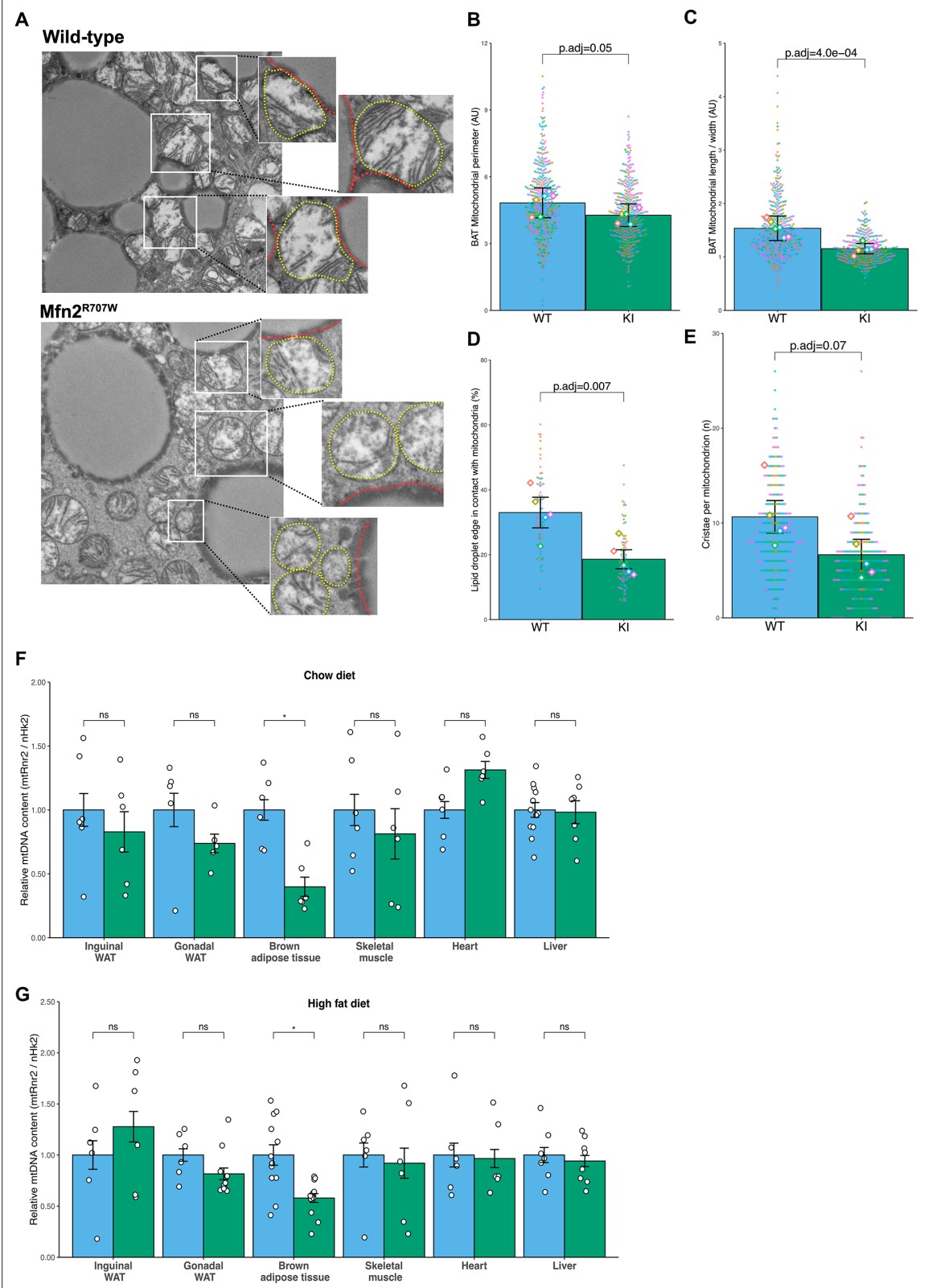

**Figure 2.** Effect of Mfn2$^{R707W}$ on mitochondrial structure and function. (**A**) Representative transmission electron microscopy (TEM) images of BAT with zoomed-in images of mitochondria (highlighted in *yellow*) bordering lipid droplets (outlined in *red*). (**B**) Quantification of mitochondrial perimeter from TEM on BAT. Each dot represents data from an individual mitochondrial cross section with each diamond showing the separate 6 biological replicates. p.adj gives the FDR-adjusted p-value from across all TEM analyses. (**C**) Quantification of mitochondrial aspect ratio (length/width) from TEM of BAT.

*Figure 2 continued on next page*

*Figure 2 continued*

(**D**) Quantification of mitochondrial-lipid droplet contact from TEM, expressed as proportion (%) of lipid droplet in contact with mitochondrial membrane on BAT. (**E**) Number of cristae per mitochondrion from TEM of BAT. (**F–G**) Mitochondrial DNA content in tissues from mice fed chow diet (**F**) or HFD (**G**) for 6 months. Each data point represents a separate animal. p-values are from pairwise comparisons (T-tests) between WT and Mfn2$^{R707W}$ KI that are FDR-adjusted for multiple tests (* p-FDR <0.05). WT in *blue*, homozygous Mfn2$^{R707W}$ KI in *green*. AU, arbitrary units.

The online version of this article includes the following source data and figure supplement(s) for figure 2:

**Figure supplement 1.** Effect of Mfn2$^{R707W}$ on mitochondrial morphology in white adipose tissue.

**Figure supplement 2.** Effect of Mfn2$^{R707W}$ on mitochondrial morphology in liver, heart, and muscle.

**Figure supplement 3.** Altered expression of Oxphos protein subunits in adipose tissue.

**Figure supplement 3—source data 1.** Raw and annotated immunoblots from *Figure 2—figure supplement 3*.

**Figure supplement 4.** Quantification of western blots for Oxphos subunits from tissues of chow-diet-fed mice.

**Figure supplement 5.** Mfn2$^{R707W}$ does not impair brown adipose tissue thermogenic capacity.

Similar results were observed in female mice harboring the Mfn2$^{R707W}$ mutation. Specifically, there was no difference in body weight, fat mass, fasting glucose or insulin between high-fat fed WT and KI mice (*Figure 3—figure supplement 2A–C*).

## Adipose-tissue specific activation of the integrated stress response in Mfn2$^{R707W}$ mice

Mitochondrial dysfunction is sensed by cells, and triggers a series of adaptive responses to maintain mitonuclear balance and cellular homeostasis (*Melber and Haynes, 2018*). Precise sensing and transducing mechanisms vary among different forms of mitochondrial perturbation (*D'Amico et al., 2017*), and show some redundancy (*Condon et al., 2021*). Although details of the integration of these mechanisms in different tissue and cellular contexts are not fully elucidated, it is clear that the transcription factor Atf4 plays a crucial role (*Lu et al., 2004*). Atf4 is translationally upregulated following phosphorylation of eIF2α, which is a point of convergence of several cellular stress sensing pathways. The canonical eIF2α kinase HRI is most closely implicated in linking mitochondrial dysfunction to eIF2α phosphorylation (*Guo et al., 2020*; *Fessler et al., 2020*), but mTORC1 appears to play a role in Atf4 upregulation independently of eIF2α phosphorylation (*Ben-Sahra et al., 2016*). The response to sustained mitochondrial dysfunction overlaps significantly with the response to many other cellular stressors including unfolded protein induced stress (i.e. the unfolded protein response [UPR]), and is best characterised as a cellular ISR (*Youle and van der Bliek, 2012*).

Strong transcriptional evidence of ISR activation was found in adipose tissue of patients with *MFN2$^{R707W}$*-related lipodystrophy (*Rocha et al., 2017*). We thus screened multiple tissues from the KI mice for ISR activation. mRNA levels of *Atf4*, *Atf5*, and *Ddit3* (Chop), all sentinel markers of the ISR, were increased in BAT and epididymal WAT of KI mice (*Figure 4A–C*). *Ddit3* was also higher in inguinal WAT from KI animals. mRNA levels of *Atf4*, *Atf5*, and *Ddit3* (Chop) were unchanged in liver, heart, and skeletal muscle, except for a 1.4-fold rise in *Atf5* (p.adj=0.03) in skeletal muscle only. Phosphorylation of eIF2α and protein expression of Mthfd2, an Atf4-upregulated enzyme playing a rate-limiting role in mitochondrial one carbon metabolism (*Shin et al., 2017*; *Forsström et al., 2019*; *Khan et al., 2017*), were also both strongly increased in BAT and in WAT in KI mice (*Figure 4D–F* and *Figure 4—figure supplement 1E–G* and *Figure 4—figure supplement 2*), whereas they were unchanged in the liver, skeletal muscle, and heart. mRNA expression of two important secreted mediators of the organismal metabolic response to mitochondrial dysfunction, Gdf15 and Fgf21, trended towards an increase in adipose tissue, but serum concentrations were unchanged in KI mice (*Figure 4—figure supplement 1A–D*). This differs from observations in patients with MFN2$^{R707W}$-related lipodystrophy (*Khan et al., 2017*; *Fisher and Maratos-Flier, 2016*; *Patel et al., 2019*), and may relate to the fact that such patients manifest non-alcoholic fatty liver disease, which was not seen in the KI animals.

Given the recent evidence implicating HRI kinase as a key mediator of eIF2α phosphorylation induced by mitochondrial dysfunction (*Guo et al., 2020*; *Fessler et al., 2020*), we next sought to determine if this pathway was active in R707W KI mice. The pathway activating HRI kinase was reported to be initiated by activation of the metalloendopeptidase Oma1 (*Baker et al., 2014*; *Anand et al., 2014*). In addition to cleaving Dele1 (*Harada et al., 2010*), which activates HRI kinase, Oma1 also cleaves Opa1 (optic atrophy type 1) (*Head et al., 2009*) and itself upon activation. We thus

**Table 1.** Weights and serum biochemistry for mice on chow or high fat diet for 6 months.

Seven-week-old mice were fed chow (n=8–12) or 45% kcal HFD (n=13–14) for 6 months. Blood was taken 4 weekly with 6 hr fasting blood taken on week 24. p-FDR are false-discovery rate adjusted p-values derived from unpaired T-tests. Values in brackets referred to standard error of the mean. ALT, alanine aminotransferase; AST, aspartate aminotransferase; FGF21, fibroblast growth factor 21; Gdf15, Growth and differentiation factor 15; HOMA-IR, Homeostatic Model Assessment for Insulin Resistance; NEFA, non-esterified fatty acids.

| | Chow diet | | | High-fat diet | | |
|---|---|---|---|---|---|---|
| | WT | Knock-in | p-FDR | WT | Knock-in | p-FDR |
| Weight (g) | 39.9 (1.2) | 40.9 (1.4) | 1 | 48.4 (1.7) | 45.7 (1.5) | 0.51 |
| Lean mass (g) | 19.97 (0.44) | 20.53 (0.53) | 0.85 | 16.50 (0.39) | 15.51 (0.42) | 0.2 |
| Fat mass (g) | 12.41 (0.97) | 12.67 (1.29) | 1 | 22.39 (1.63) | 24.09 (1.10) | 0.79 |
| Leptin (ug/L) | | | | | | |
| 4 weeks | 3.6 (0.9) | 1.3 (0.2) | 0.17 | 14.3 (2.6) | 4.8 (0.5) | 0.01 |
| 8 weeks | 4.5 (0.8) | 1.6 (0.3) | 0.056 | 20.8 (3.3) | 8.6 (0.9) | 0.007 |
| 12 weeks | 7.6 (0.4) | 2.3 (0.5) | 0.045 | 19.3 (4.0) | 11.4 (1.5) | 0.31 |
| 24 weeks (fasting) | 15.4 (3.4) | 6.0 (1.3) | 0.047 | 57.0 (8.9) | 15.6 (1.5) | 0.003 |
| Adiponectin (mg/L) | | | | | | |
| 4 weeks | 24.8 (1.0) | 16.4 (2.3) | 0.068 | 21.0 (0.4) | 11.0 (3.3) | 4.20E-10 |
| 8 weeks | 20.0 (0.7) | 11.0 (0.7) | 1.50E-05 | 29.4 (1.4) | 16.0 (0.7) | 5.30E-07 |
| 12 weeks | 29.0 (1.4) | 14.7 (0.7) | 4.50E-05 | 33.0 (0.8) | 18.2 (0.9) | 7.20E-09 |
| 24 weeks (fasting) | 29.0 (2.3) | 15.4 (2.3) | 0 | 27.9 (0.9) | 15.5 (0.6) | 4.30E-09 |
| Fgf21 (ng/L) | | | | | | |
| 20 weeks | 238 (26) | 292 (22) | 0.50 | 2,655 (203) | 2,475 (268) | 1 |
| 28 weeks | 444 (51) | 638 (96) | 0.24 | 3,355 (911) | 2,919 (289) | 1 |
| Gdf15 (ng/L) (28 weeks) | 132 (6) | 172 (14) | 0.067 | 250 (36) | 217 (23) | 1 |
| Glucose (mmol/L) | 10.0 (0.5) | 10.4 (0.4) | 1 | 11.7 (0.5) | 11.4 (0.5) | 1 |
| Insulin (μg/L) (33 weeks) | 1.8 (0.3) | 2.0 (0.3) | 1 | 1.7 (0.2) | 1.2 (0.1) | 0.17 |
| HOMA-IR | 141.3 (29.0) | 159.2 (23.3) | 1 | 152.2 (22.9) | 104.8 (12.1) | 0.26 |
| Insulin (μg/L) (GTT baseline) | 1.8 (0.4) | 2.3 (0.7) | 1 | 2.3 (0.4) | 1.4 (0.1) | 0.1 |
| Lactate (mmol/L) | 4.2 (0.3) | 4.4 (0.2) | 1 | 4.5 (0.4) | 4.8 (0.2) | 1 |
| NEFA (μmol/L) | 1,688 (78) | 1,564 (46) | 0.39 | 922 (87) | 1030 (90) | 0.90 |
| TG (mmol/L) | 1.1 (0.1) | 1.0 (0.1) | 1 | 0.8 (0.1) | 0.7 (0.0) | 0.76 |
| Total cholesterol (mmol/L) | 3.0 (0.1) | 3.1 (0.1) | 1 | 6.3 (0.3) | 5.2 (0.2) | 0.042 |
| ALT (IU/L) | 64 (6) | 68 (5) | 1 | 144 (33) | 45 (9) | 0.076 |
| AST (IU/L) | 114 (20) | 118 (16) | 1 | 207 (32) | 162 (22) | 0.65 |

immunoblotted tissue lysates for Opa1 and Oma1. No evidence for Oma1 activity was found in liver, skeletal muscle or heart (*Figure 4—figure supplement 3*). In BAT, we did detect a modest change in Oma1 expression and a slight increase in the short / long Opa1 ratio, providing some evidence that the pathway may be activated (*Figure 4—figure supplement 4*). However, the differences observed were small precluding a confident conclusion at this stage. In WAT, clear bands for all five Opa1 forms could not be visualised on repeated attempts and so we did not attempt to quantify the data. Current WAT data thus do not suggest Oma1 activation (*Figure 4—figure supplement 4C–D*). As far as we are aware, Oma1 activation has yet to be examined in Mfn2 tissue-specific KO models so we cannot infer anything from those studies either. Thus we conclude that the ISR is activated in selected tissues in the KI mice, but the question of how this is linked to the Mfn2 R707W mutation remains open.

To obtain an unbiased view of the transcriptional consequences of the lipodystrophy-associated Mfn2 mutation, we next applied bulk RNA sequencing (RNAseq) to BAT and inguinal WAT from

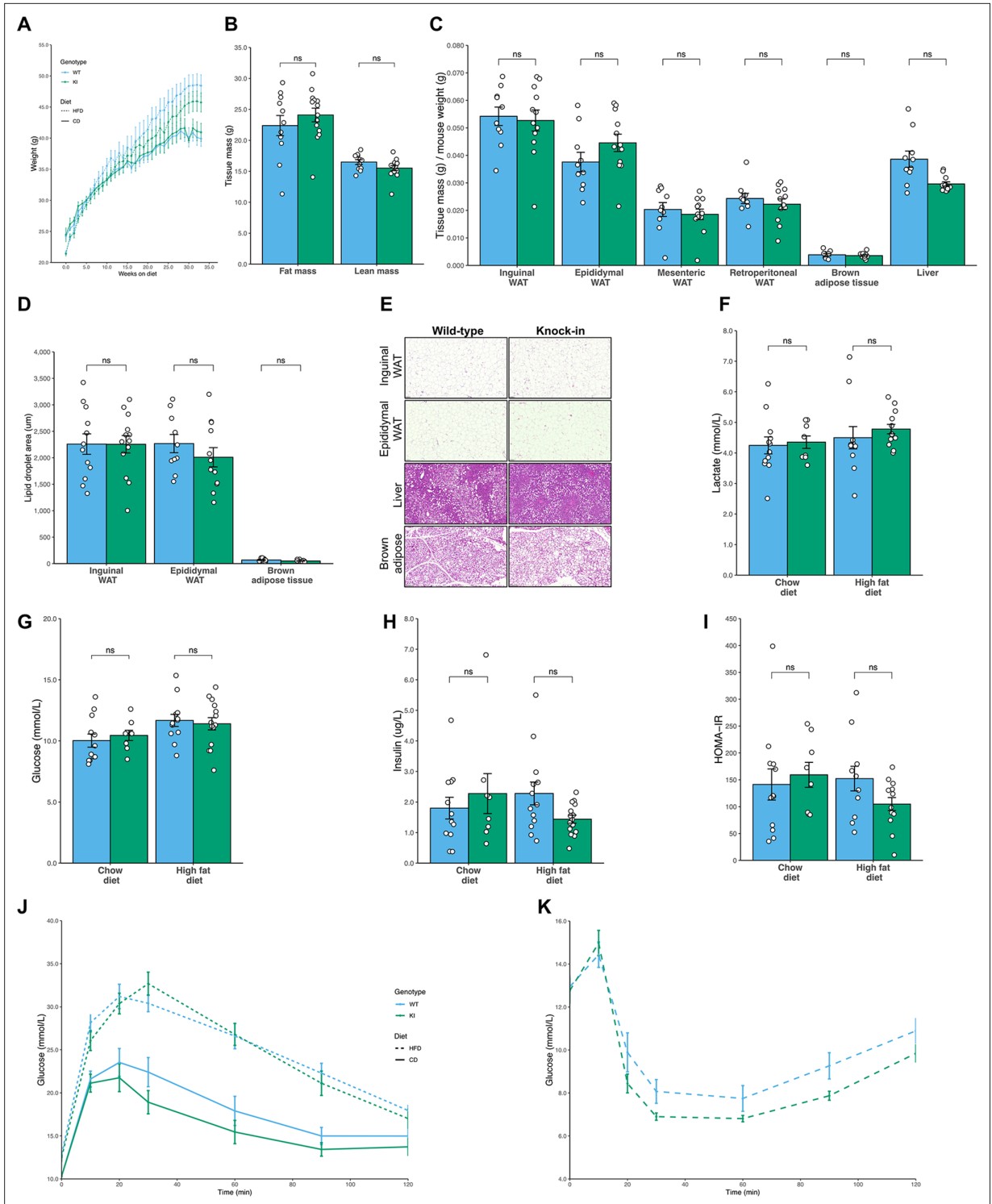

**Figure 3.** No difference in fat mass or glucose homeostasis in Mfn2$^{R707W}$ mice compared to WT mice on chow or high fat diet. Seven-week-old mice were fed chow (n=8–12) or 45% kcal HFD (n=13–14) for 6 months. (**A**) Absolute body mass for mice fed chow (solid line) and HFD (dashed line) over 6 months. (**B**) Time-domain nuclear magnetic resonance (TD-NMR) measurement of fat and lean mass of mice fed HFD. (**C**) The ratio of tissue weights versus body mass for multiple tissues, including four WAT depots, from mice fed HFD. (**D**) Quantification of lipid droplet area from histological specimens of adipose tissue from mice fed HFD for 6 months. (**E**) Representative histological images from WAT, liver, and BAT from mice fed HFD. Analyses of mouse plasma lactate (**F**), plasma glucose (**G**), and insulin (**H**), homeostatic model of assessment of insulin resistance (HOMA-IR, **I**) after a 6 hr fast. Change in plasma glucose during intraperitoneal glucose tolerance test (**J**) and intraperitoneal insulin tolerance test (**K**). ns, p>0.05 on unpaired T-tests, adjusted for multiple comparisons. WT in *blue*, homozygous Mfn2$^{R707W}$ KI in *green*. Each data point represents an individual animal.

*Figure 3 continued on next page*

*Figure 3 continued*

The online version of this article includes the following figure supplement(s) for figure 3:

**Figure supplement 1.** No evidence of altered fat mass or glucose homeostasis in Mfn2$^{R707W}$ compared to WT mice on chow or high fat diet for 6 months.

**Figure supplement 2.** Similar phenotype in female Mfn2$^{R707W}$ knock-in mice fed high fat diet for 6 months.

HFD-fed mice. Inguinal, rather than epididymal, WAT was selected as subcutaneous adipose tissue is predominantly affected in human MFN2-related and other partial lipodystrophies (*Mann and Savage, 2019*). Induction of the ISR was confirmed in both BAT and WAT (*Figure 4G–I* and *Figure 4—figure supplement 5*), with the "unfolded protein response" gene set, a surrogate for the ISR, the top upregulated gene set in BAT and 6th in inguinal WAT. *Atf5* and *Mthfd2* were confirmed among the most highly upregulated mRNAs in both tissues, and a range of other well established ISR genes also showed increased expression. These included *Ddit3* (Chop), *Trib3*, an Atf4-driven ISR component that exerts negative feedback on the ISR, and *Gadd45a*, involved in ISR-induced G2/M checkpoint arrest (*Lee et al., 2019*).

The top upregulated gene set in inguinal WAT was 'oxidative phosphorylation', driven solely by increased expression of nuclear-encoded mitochondrial genes (*Figure 4—figure supplement 1H*). In contrast, mitochondria-encoded genes were nearly universally downregulated (*Figure 4—figure supplement 1I*), recapitulating the pattern seen in affected human WAT (*Rocha et al., 2017*). In BAT, a similar but weaker pattern was seen on inspection of heatmaps, but this was not sufficient to reach statistical significance.

Another finding common to mouse and human was the transcriptional evidence of mTORC1 activation. The 'mTorc signalling' gene set was the second most upregulated group in BAT and third most upregulated in WAT (*Figure 4—figure supplement 5B, E*). This activation is consistent with the proposed role for mTORC1 in mediating the proximal ISR (*Ben-Sahra et al., 2016*), and is of interest given accumulating evidence that the mTORC1 inhibitor sirolimus may exert beneficial effects in various mitochondrial diseases (*Johnson et al., 2013*).

Although no increase in adipose tissue mass was seen in *Mfn2$^{R707W}$* mice the 'adipogenesis' gene set was upregulated in inguinal WAT under both diet conditions (*Figure 4H*, *Figure 4—figure supplement 5B*). However, closer inspection revealed a mixed profile of individual gene expression. The most consistent finding was downregulation of the adipokine-encoding mature adipocyte genes *Adipoq* and *Lep* in *Mfn2$^{R707W}$* mice (discussed below). *Adipoq* and *Lep* were also downregulated in BAT (fold changes 0.64 (p.adj=6.3 × 10$^{-4}$) *and* 0.31 (p.adj=0.03) respectively), but the adipogenesis gene set was not enriched overall (*Figure 4—figure supplement 5E*).

To assess for other potential drivers of adipose hyperplasia, we also examined downregulated gene sets. The signature of epithelial-mesenchymal transition (EMT) was the most strongly downregulated set in mouse BAT and WAT (*Figure 4I*, *Figure 4—figure supplement 5C, F*), and was also previously found to be downregulated in overgrown human WAT in MFN2-associated multiple symmetric lipomatosis (*Rocha et al., 2017*). In bulk transcriptomic data it is not possible to discern the cell type(s) responsible for this consistent signature. However, TGFβ family ligands are important mediators of EMT, some family members inhibit adipogenesis (*Zamani and Brown, 2011*), and they also play important roles in regulating mitochondrial function and in responding to mitochondrial dysfunction (*Casalena et al., 2012*).

We next sought to assess whether the increased demand for adipose expansion imposed by HFD feeding exacerbates the transcriptional consequences of Mfn2 R707W homozygosity. RNAseq was thus undertaken of inguinal WAT from mice maintained on HFD for 6 months. Comparison to WT animals revealed strikingly concordant findings to those seen in chow-fed mice (*Figure 4—figure supplement 5G*). No general differences were seen in the magnitude of transcriptional changes induced by Mfn2 R707W homozygosity between conditions. Oxidative phosphorylation, unfolded protein response and adipogenesis Hallmark gene sets were strongly upregulated, whereas genes in the EMT set were downregulated in KI mice on both diets. An exception was the group of mRNAs related to cholesterol homeostasis, for which diet strikingly modified the effect of genotype. They were the top downregulated gene set in inguinal WAT in HFD-fed animals but were not significantly altered on chow in the same depot and were upregulated in BAT (*Figure 4I*, *Figure 4—figure supplement 5E*). Lower expression of key enzymes in cholesterol metabolism (e.g. Hydroxymethylglutaryl-CoA

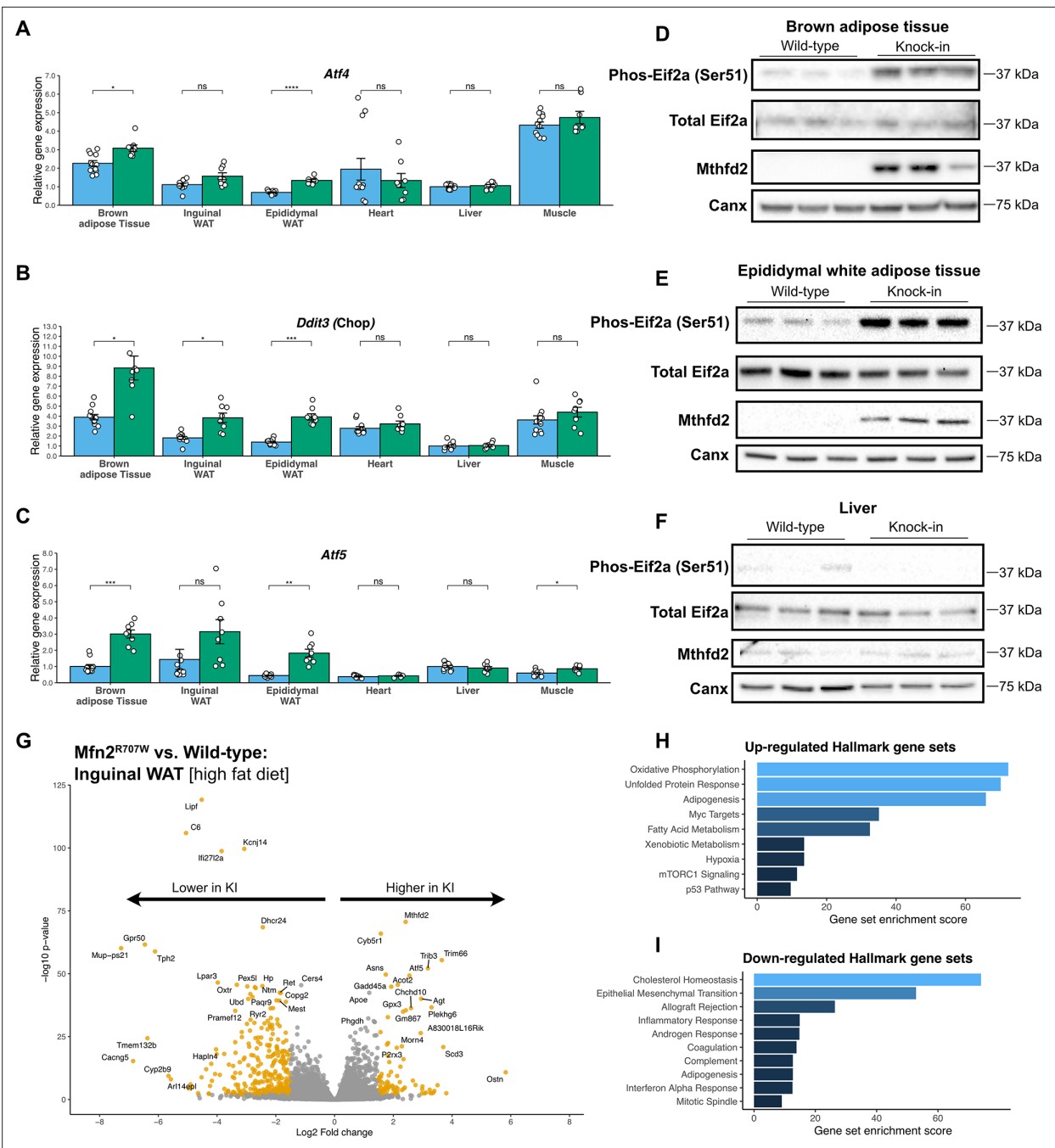

**Figure 4.** Mfn2[R707W] causes an adipose-tissue-specific induction of the integrated stress response. qPCR of genes involved in the integrated stress response (ISR) for six tissues from animals fed chow diet for 6 months: *Atf4* (**A**), *Ddit3* (Chop, **B**), and *Atf5* (**C**). Each data point represents one animal. Target gene CT values were normalised to three housekeeping genes (*36b4, B2m,* and *Hprt*) and expressed relative to WT liver for each gene. WT in *blue*, homozygous Mfn2[R707W] in *green*. p-values are FDR-adjusted for multiple tests. Western blots from BAT (**D**), epididymal WAT (**E**), and liver (**F**) illustrating Ser51-phosphorylation of eIF2α and expression of Mthfd2 with calnexin (*Canx*) as loading control. Western blots are representative of at least three biological and technical replicates. (**G**) Volcano plot from bulk RNA sequencing (n=8 per genotype) of inguinal WAT from mice on HFD. Significantly differentially expressed genes (Log$_2$ fold change >1.5 and p-FDR <0.001) are highlighted in orange. Pathway analysis using significantly differentially expressed genes for upregulated (**H**) and downregulated (**I**) Hallmark gene sets. The X-axis depicts a relative gene set enrichment score. All illustrated gene sets are enriched with p-FDR <0.05. Asterisks indicate p-values from pairwise comparisons (T-tests) between WT and Mfn2[R707W] KI that are FDR-adjusted for multiple tests (* p-FDR <0.05, ** p-FDR <0.01, *** p-FDR <0.001, **** p-FDR <0.0001).

The online version of this article includes the following source data and figure supplement(s) for figure 4:

**Source data 1.** Raw and annotated immunoblots from *Figure 4D–F*.

*Figure 4 continued on next page*

*Figure 4 continued*

**Figure supplement 1.** Mfn2[R707W] causes adipose tissue-specific induction of the integrated stress response with perturbation of mitochondrial gene expression.

**Figure supplement 1—source data 1.** Raw and annotated immunoblots from *Figure 4—figure supplement 1E–G*.

**Figure supplement 2.** Quantification of western blots for components of the integrated stress response in high-fat diet-fed mice.

**Figure supplement 3.** Mfn2[R707W] does not affect Oma1 or Opa1 processing in heart, liver, and muscle.

**Figure supplement 3—source data 1.** Raw and annotated immunoblots from *Figure 4—figure supplement 3*.

**Figure supplement 4.** Evidence of modest activation of the Oma1-Opa1 pathway in brown adipose tissue from Mfn2[R707W] mice.

**Figure supplement 4—source data 1.** Raw and annotated immunoblots from *Figure 4—figure supplement 4*.

**Figure supplement 5.** Transcriptional evidence of upregulation of the unfolded protein response and mTorc1 pathways in adipose tissue.

synthase [*Hmgcs1*], mevalonate kinase [*Mvk*], and squalene monooxygenase [*Sqle*]) in WAT on HFD is consistent with the response to inhibition of the mitochondrial respiratory chain in primary human fibroblasts (*Wall et al., 2022*).

## Lower circulating leptin and adiponectin in mice homozygous for Mfn2[R707W]

One of the most striking aspects of the Mfn2[R707W]-associated lipodystrophy phenotype is the low or undetectable serum leptin concentration despite abundant whole body adiposity, accounted for mostly by excess upper body adipose tissue of relatively normal histological appearance (*Rocha et al., 2017*). Serum adiponectin concentrations are also low, however this is in keeping with the 'adiponectin paradox' widely seen in obesity with insulin resistance (*Arita et al., 1999*). Mirroring these human observations, KI mice showed low-serum leptin and adiponectin concentrations on both chow and HFD (*Figure 5A–C*), although unlike humans, the mice had normal fat mass and insulin sensitivity. In both WT and KI mice, serum leptin concentrations correlated positively with whole body adiposity on chow and HFD, but a generalised linear model revealed marked attenuation of the relationship between serum leptin concentration and adipose mass in the KI mice (*Figure 5C*, *Figure 5—figure supplement 1A*). Immunoblotting of WAT samples from KI mice fed a HFD confirmed that both leptin and adiponectin expression was reduced (*Figure 5—figure supplement 2*). Findings were similar in female mice, in which serum adiponectin was significantly lower in KI mice (*Figure 3—figure supplement 2F*), with a trend towards lower leptin (*Figure 3—figure supplement 2G–I*).

RNAseq showed *Lep* mRNA in inguinal WAT to be lower in Mfn2[R707W] than in WT mice under both chow (fold change 0.35; p.adj=$3.2 \times 10^{-4}$, *Figure 4—figure supplement 5A*) and HFD (fold change 0.66; p.adj=$5.3 \times 10^{-4}$, *Figure 4G*). *Adipoq* mRNA was also significantly lower in chow (fold change 0.66; p.adj=$2.7 \times 10^{-4}$, *Figure 4—figure supplement 5A*) and HF-fed mice (fold change 0.57; p.adj=$3.3 \times 10^{-15}$, *Figure 4G*). RT-qPCR analysis confirmed lower *Lep* mRNA in epididymal WAT from chow fed Mfn2[R707W] animals (*Figure 5D*). RT-qPCR analysis also found 50% lower *Adipoq* mRNA in both inguinal and epididymal WAT in chow fed Mfn2[R707W] mice and in inguinal WAT from HFD fed Mfn2[R707W] mice (*Figure 5E*).

To assess leptin secretion directly, we studied production of adipokines from adipose explants. Explants from KI mice fed on HFD for 4 weeks showed lower secretion of leptin and adiponectin per gram of tissue (*Figure 5—figure supplement 1A–B*). KI explants also exhibited minimal increase in leptin secretion after insulin and dexamethasone stimulation. *Adipoq* mRNA was lower in KI than WT explants at baseline whereas *Lep* mRNA was no different at baseline but failed to increase in KI explants following insulin and dexamethasone stimulation (*Figure 5—figure supplement 1C*).

To assess whether induction of the ISR in adipose tissue may be responsible for the relative leptin deficiency in both humans and mice homozygous for MFN2 R707W, we studied adipocytes freshly isolated from mouse gonadal fat in floating culture, adapting a recently described protocol (*Li et al., 2021*). We induced the ISR using each of two different well characterised activators, namely thapsigargin (TG), an inhibitor of the endoplasmic reticulum (ER) $Ca^{2+}$ ATPase that depletes ER calcium (*Wictome et al., 1992*; *Harding et al., 2000*), or tunicamycin (TN), which blocks protein glycosylation. ISR induction was confirmed by increased *Atf4* and *Ddit3* mRNA and/or protein expression, and by eIF2α phosphorylation (*Figure 5F–L*). *Lep* mRNA was modestly reduced by TG but not by

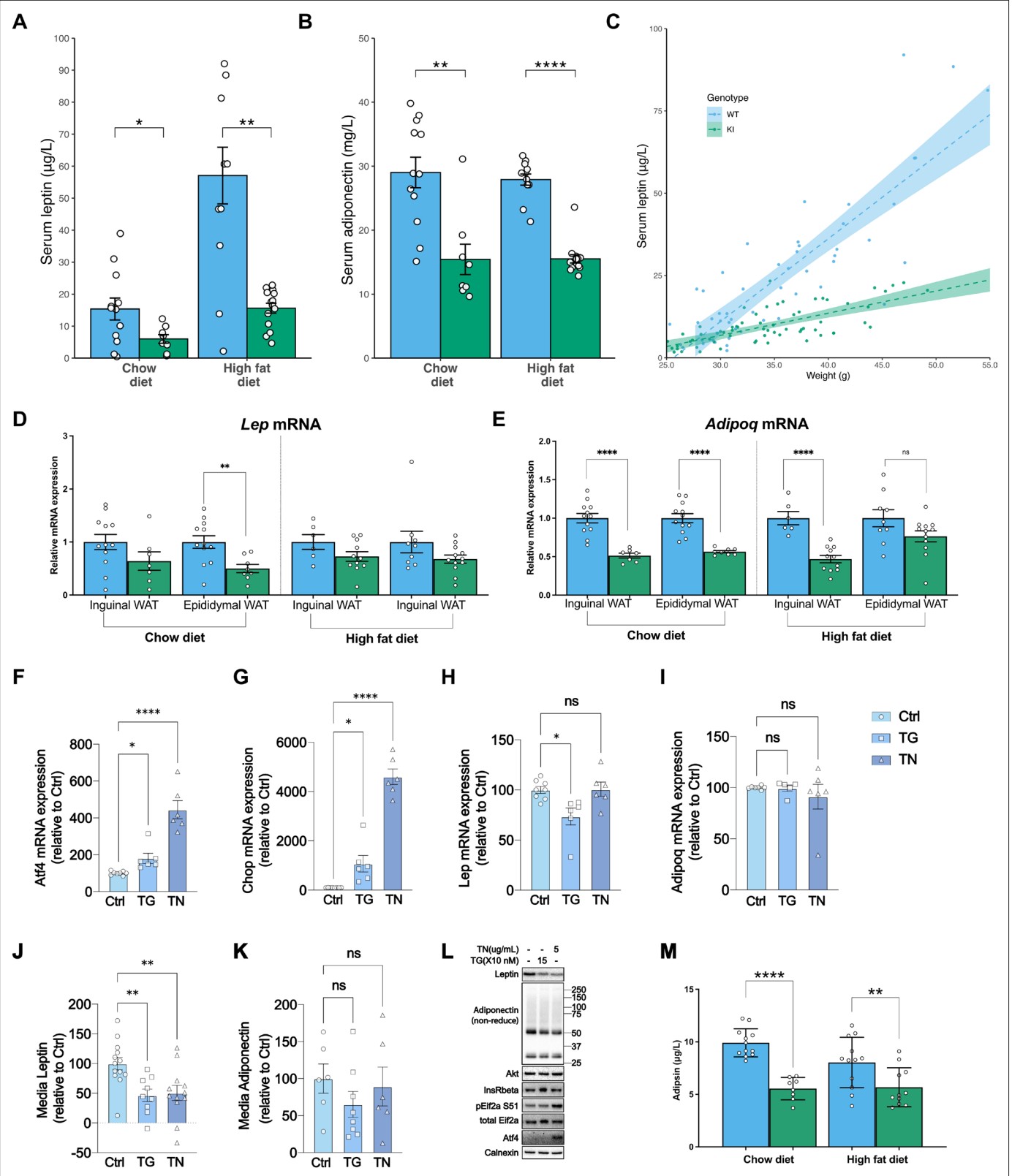

**Figure 5.** Mfn2R707W decreases adipose secretion of leptin and adiponectin. Fasting serum leptin (**A**) and adiponectin (**B**) from mice after 6 months on chow diet or HFD. Asterisks indicate t-tests comparing WT and Mfn2R707W KI with p-values adjusted for multiple testing. (**C**) Relationship between leptin and body weight for animals fed HFD. Each data point represents one measurement of leptin, with multiple measurements per animal. Shaded area represents the 95% confidence interval. Data are from n=13–14 animals. qPCR of *Lep* (**D**) and *Adipoq* (**E**) from WAT depots. Each data point represents

*Figure 5 continued on next page*

*Figure 5 continued*

data from a separate animal. Target gene CT values were normalised against three housekeeping genes (*36b4, B2m,* and *Hprt*) and expressed relative to WT for each condition. (**F–I**) mRNA expression of *Atf4, Chop, Lep,* and *Adipoq* in primary adipocytes treated with either DMSO control or Thapsigargin (TG, 150 nM) or Tunicamycin (TN, 5 µg/mL) for 6 hr. Each data point represents an individual well from a separate biological experiment (n=5). Asterisks indicate significance on one-way ANOVA with correction for multiple comparisons. (**J**) Leptin and (**K**) Adiponectin secretion from primary adipocytes treated with either DMSO control or Thapsigargin (TG, 150 nM) or Tunicamycin (TN, 5 µg/mL) over 6 hr. Asterisks indicate significance on one-way ANOVA with correction for multiple comparisons. (**L**) Representative western blots of primary adipocytes treated with either DMSO control or Thapsigargin (TG, 150 nM) or Tunicamycin (TN, 5 µg/mL) for 6 hr. (**M**) Fasting serum adipsin (complement factor D) from mice after 6 months on chow diet or HFD. Asterisks indicate p-values from pairwise comparisons (T-tests) between indicated groups that are FDR-adjusted for multiple tests (* p-FDR <0.05, ** p-FDR <0.01, *** p-FDR <0.001, **** p-FDR <0.0001).

The online version of this article includes the following source data and figure supplement(s) for figure 5:

**Source data 1.** Raw and annotated immunoblots from *Figure 5L*.

**Figure supplement 1.** Mfn2$^{R707W}$ decreases adipose leptin and adiponectin secretion from multiple adipose depots and in different dietary conditions.

**Figure supplement 2.** Mfn2$^{R707W}$ decreases leptin and adiponectin protein expression in white adipose tissue.

TN (*Figure 5H*), whereas both agents reduced intracellular leptin protein expression (*Figure 5L*) and secretion (*Figure 5J*). Expression of adiponectin was also reduced without a change in mRNA level whereas expression of Akt and the insulin receptor beta subunit were not altered (*Figure 5I and K & L*). These data suggest that ISR activation may have a bigger impact on secreted proteins than on intracellular proteins, in keeping with the previous suggestion that the ISR tends to prevent a fall in intracellular amino acid concentrations (*Harding et al., 2000*). In seeking to validate this notion, we proceeded to measure the serum concentration of adipsin, another 'adipokine' selectively secreted by adipocytes. Serum adipsin concentrations were significantly lower in the KI mice than in WT controls in both chow- and HFD-fed mice (*Figure 5M*), suggesting that low serum leptin and adiponectin may be part of a wider pattern of reduced adipocyte-derived secreted proteins in *MFN2$^{R707W}$* KI mice.

## Discussion

The recent discovery that humans homozygous for the MFN2 R707W mutation manifest striking adipose redistribution associated with serious metabolic disease is probably the clearest example to date of a causal link in humans between a mitochondrial perturbation and adipose dysregulation. MFN2-related lipodystrophy has some remarkable and currently poorly understood features. These include: (a) a marked and often dramatic increase in upper body adiposity, contrasting with loss of lower limb adipose tissue; (b) a severe reduction in plasma leptin concentration despite abundant, histologically near-normal upper body fat. These problems have to date been associated only with the R707W mutation. To investigate the molecular pathogenesis of MFN2 R707W-related lipodystrophy, and the role of MFN2 in leptin synthesis and secretion, we generated and characterised homozygous *Mfn2$^{R707W/R707W}$* mice.

*Mfn2* knock-out mice die in early embryogenesis (*Chen et al., 2003*) while mice homozygous for either of two human neuropathy-associated, GTPase null missense mutations (H361Y or R94W) die on day 0–1 (*Strickland et al., 2014*). Mfn2 is nearly ubiquitously co-expressed with its closely related paralogue Mfn1, and this demonstrates that it has essential, non-redundant functions. Homozygous Mfn2 R707W mice, in contrast, were viable and bred normally, showing that Mfn2 R707W retains significant function. Mfn2 also has key metabolic functions in mature adipocytes: mice lacking Mfn2 in all adipocytes (*Boutant et al., 2017*) or in brown adipocytes alone (*Mahdaviani et al., 2017*) did not show reduced adipose tissue, but they did exhibit lower energy expenditure, reduced expression of multiple oxidative phosphorylation subunits, and impaired cold tolerance. Paradoxically, however, both lines were protected from systemic insulin resistance. Mice in which Mfn2 was deleted in all adipocytes in adulthood showed increased obesity and elevated blood glucose (*Mancini et al., 2019*). In contrast, homozygous Mfn2 R707W mice showed no overt change in adipose mass, metabolic function, or thermogenic capacity even though the genetic alteration was constitutional. This confirms some retained Mfn2 function also in adipose lineages.

Primary anatomical and/or functional defects in humans with MFN2 R707W homozygosity have been observed only in adipose tissue and peripheral nerves, with some but not all people reported to have sensorimotor neuropathy. Such neuropathy is commonly observed in people with heterozygous

loss of MFN2 function (*Chung et al., 2006*; *Pipis et al., 2020*). Although homozygous Mfn2 R707W KI mice had no overt anatomical adipose abnormality, and failed to show obvious neurological phenotypes, ultrastructural studies did reveal mitochondrial network disruption in adipose tissues, but not liver, skeletal muscle, or heart. The structural changes in mouse adipose mitochondria resembled those in adipose tissue from patients homozygous for the MFN2 R707W mutation (*Rocha et al., 2017*), and in both species these were associated with robust activation of the ISR, which was also seen previously in tissue-specific *Mfn2* knock-out mice (*Sebastián et al., 2012*). The ISR was not activated in liver, muscle, and heart, strengthening evidence that Mfn2 R707W has deleterious effects selectively in adipose tissue. We cannot conclusively exclude the possibility that tissues other than brown and white adipose tissue are affected, as we have not studied every tissue in the mice or patients, but if present, it is not associated with overt phenotypes.

The reason for adipose-selectivity of abnormalities in Mfn2 R707W homozygous mice is not established, but disruption of the function of Mfn2 in establishment or maintenance of mitochondrial-lipid droplet contact sites, perhaps through interaction with an adipose-specific protein, is plausible. We did observe reduced mitochondrial-lipid droplet contacts in brown adipose tissue from Mfn2 R707W KI mice, as reported in adipocyte *Mfn2* knock-out animals, but we were unable to replicate the direct mitofusin 2-perilipin 1 interaction previously reported using co-immunoprecipitation (*Boutant et al., 2017*). This requires further characterisation in the context of Mfn2$^{R707W}$.

Whether mitochondrial structural and functional perturbation mediates the overgrowth of some adipose depots and loss of others in humans with MFN2 R707W homozygosity remains to be proven. If it does, the mechanisms transducing dysfunction of a key organelle into cellular hyperplasia in some adipose depots but loss of adipose tissue in others are also unexplained. KI mice exhibit neither adipose loss nor hyperplasia, even when challenged by a HFD. This failure to model the gross anatomical adipose abnormalities of humans, despite evidence of mitochondrial dysfunction and attendant ISR, establishes that the cellular abnormalities we describe are not sufficient to perturb adipose growth, but they may still be necessary. Whether a permissive genetic background, or an undefined additional stressor, are required as cofactors, remains to be determined.

Some of the transcriptomic changes observed that are common to mouse and human adipose tissue do suggest both potential opportunities to intervene pharmacologically, and mechanistic hypotheses relating to adipose hyperplasia that warrant further investigation. For example, transcriptional evidence of strong mTORC1 activation, likely part of the proximal ISR triggered by mitochondrial dysfunction, suggests that mTOR inhibitors such as sirolimus are worthy of testing. It is possible that they may reduce the ISR, thereby restraining compensatory adipose hyperplasia or even inducing synthetic lethality in cases of MFN2 R707W homozygosity. Several previous studies have suggested that mTOR inhibition may have beneficial effects in other primary mitochondrial disorders (*Johnson et al., 2013*; *Civiletto et al., 2018*). Downregulation of TGFβ also merits further investigation as one candidate mechanism linking Mfn2 R707W homozygosity to adipose hyperplasia. This is based on strong transcriptional evidence of downregulated EMT in both mice and humans, on the important roles of TGFβ in EMT and adipogenesis (*Ishay-Ronen et al., 2019*; *Battula et al., 2010*), and on the inter-relationship of TGFβ signalling with mitochondrial dysfunction (*Dimeloe et al., 2019*; *Patel et al., 2015*; *Sun et al., 2019*; *Yi et al., 2015*).

A further notable difference between the mice and humans with the homozygous MFN2 R707W mutation is that serum concentrations of GDF15 and FGF21 were increased in people but not mice (*Table 2*). This likely reflects the fact that affected humans also have fatty liver disease and diabetes, both strongly associated with elevated stress hormone levels (*Vila et al., 2011*; *Koo et al., 2018*). In keeping with this, we have shown in mice that the liver is the predominant source of circulating FGF21 and GDF15, with little or no contribution from adipose tissue (*Patel et al., 2022*).

Although abnormal adipose growth and metabolic disease were not seen in KI mice, the low plasma leptin concentration seen in human adipose overgrowth associated with MFN2 mutations was replicated. Leptin concentrations were not reported in previously described adipocyte *Mfn2* knock-out mice (*Boutant et al., 2017*; *Mancini et al., 2019*), and although a different model of adipose-specific mitochondrial dysfunction (*Tfam* knock-out) did show reduced serum leptin, fat mass was also reduced compared to WT (*Vernochet et al., 2012*). Lower adipose leptin secretion caused by Mfn2 R707W does not appear to be predominantly transcriptionally mediated in mice, as in some analyses, for example of adipose explants under basal conditions, leptin secretion was reduced

**Table 2.** Comparison of human *MFN2*^R707W-associated lipodystrophy with phenotype of *Mfn2*^R707W/R7007W mice.

BAT, brown adipose tissue; GTT, glucose tolerance test; HFD, high-fat diet; ISR, integrated stress response; MFN2, mitofusin 2; mRNA, messenger ribose nucleic acid; mtDNA, mitochondrial DNA; PLIN1, perilipin 1; TEM, transmission electron microscope; WAT, white adipose tissue.

| Human | Phenocopy? | Mouse |
|---|---|---|
| Upper body adipose overgrowth | No | No difference in weight of any adipose depots |
| Lower limb lipoatrophy | No | |
| Insulin resistance | No | No difference on GTT or fasting insulin |
| Lower serum leptin concentration | Yes | Seen on chow or HFD; lower secretion seen from adipose explants |
| Severely reduced adipose leptin mRNA expression | Partial | Modest decrease only; only significant in some analyses |
| Lower serum adiponectin concentration | Yes | Seen on chow or HFD; lower secretion seen from adipose explants |
| Severely reduced adipose adiponectin mRNA expression | Yes | Seen on chow or HFD and in adipose explants |
| No change in WAT MFN2 protein expression (only over-grown WAT studied) | Yes | No difference in any tissue studied, though variable in WAT and BAT. |
| No difference in adipocyte size | Yes | True on both chow and HFD |
| Disorganised, fragmented WAT mitochondria on TEM | Partial | More circular mitochondria with trend towards reduced cristae |
| Upregulation of nuclear and down regulation of mitochondrial Oxphos transcriptional pathway | Yes | True in inguinal and epididymal WAT, and BAT |
| Lower Oxphos complex II and III but preserved complex I and IV protein | Partial | Lower complex I and IV protein in WAT |
| Lower WAT mtDNA | Partial | Lower mtDNA in BAT but not WAT |
| Transcriptional activation of ISR | Yes | Seen in inguinal and epididymal WAT, and BAT |

(*Figure 5—figure supplement 1A*) without alteration of leptin mRNA (*Figure 5—figure supplement 1C*). Our findings suggest instead that the lower leptin secretion is a consequence of ISR activation. Synthesis of adipokines is an amino acid-intensive process, and activation of the UPR typically results in conservation of amino acids, in part through reduction of protein secretion (*Harding et al., 2000*; *Cortopassi et al., 2006*; *Kilberg et al., 2009*; *Hetz et al., 2020*). Stressing primary adipocytes with tunicamycin or thapsigargin reduced leptin secretion without any effect on expression of non-secreted proteins such as the insulin receptor. We also observed upregulation of pathways related to amino acid metabolism (particularly in BAT) (*Figure 4—figure supplement 4*), which would be consistent with the known transcriptional effects of Atf4 (*Harding, 2003*). This suggests that the low leptin may not be due to a mitofusin-specific mechanism, rather secondary to activation of the ISR in adipose tissue as part of amino acid conservation. KI mouse data suggest that other adipokines, including adiponectin and adipsin, are similarly affected.

This study has limitations. We only characterised male homozygous KI mice in detail, so cannot extrapolate our results to females with confidence, though the limited analyses we did do in female mice were broadly consistent with the data from male mice and, case series do not suggest significant sexual dimorphism in the human disorder (*Rocha et al., 2017*; *Sawyer et al., 2015*; *Capel et al., 2018*). We also did not study heterozygous animals, but as human MFN2 R707W-associated lipodystrophy shows recessive inheritance, and as even homozygous mice do not exhibit lipodystrophy, a phenotype in heterozygous animals seems unlikely. It is unclear to what extent the phenotype observed in BAT is

due to reduction in expression of both the mitofusins. Given the concomitant reduction in expression of Oxphos components in BAT, the lower mitofusin expression may be more reflective of general mitochondrial perturbation. Lastly, this study has not directly assessed the ability of Mfn2$^{R707W}$ mutants to mediate mitochondrial fusion. However given the normal mitochondrial network morphology in non-adipose tissues in homozygous KI mice and in dermal fibroblasts from humans homozygous for MFN2 R707W (*Rocha et al., 2017*), any defect is likely mild and context dependent.

## Conclusion

Mfn2$^{R707W}$ KI mice show adipose-selective alteration of mitochondrial morphology and robust activation of the integrated stress response, but no abnormal adipose growth or systemic metabolic derangement. The KI mice do show suppressed leptin expression and plasma leptin concentration, likely secondary to the adipose selective mitochondrial stress response. The unique association of human lipodystrophy with the MFN2 R707W allele remains unexplained, but transcriptomic analysis suggested that reduced TGFβ signalling warrants further explanation as a potential cause of adipose hyperplasia, while mTOR inhibitors are worthy of testing in models as a potential targeted therapy.

## Methods

**Key resources table**

| Reagent type (species) or resource | Designation | Source or reference | Identifiers | Additional information |
|---|---|---|---|---|
| Strain, strain background (*Mus musculus, C57BL/6 J*) | Mfn2$^{R707W}$ | This paper | | See "Generation of the Mfn2$^{R707W/R707W}$ knock-in mouse" |
| Antibody | Mouse monoclonal anti-Adiponectin | GeneTex | Cat# GTX80683 | WB (1:1000) |
| Antibody | Rabbit polyclonal anti-Leptin | Abcam | Cat# ab9749 | WB (1:1000) |
| Antibody | Mouse monoclonal anti-Mfn1 | Abcam | Cat# Ab126575 | WB (1:250) |
| Antibody | Rabbit monoclonal anti-Mfn2 | Cell signalling | Cat# D2D10 | WB (1:1000) |
| Antibody | Rabbit polyclonal anti-Mthfd2 | Proteintech | Cat# 12270–1-AP | WB (1:1000) |
| Antibody | Rabbit polyclonal anti-Oma1 | Proteintech | Cat# 17116–1-AP | WB (1:1000) |
| Antibody | Mouse monoclonal anti-Opa1 | BD Biosciences | Cat# 612606 | WB (1:1000) |
| Antibody | Mouse monoclonal OXPHOS cocktail | Abcam | Cat# Ab110413 | WB (1:1000) |
| Antibody | Rabbit monoclonal anti-Phos-Eif2a | Epitomics | Cat# 10901 | WB (1:1000) |

A full list of antibodies and primers are available in *Supplementary files 1 and 2*.

## Generation of the *Mfn2$^{R707W/R707W}$* knock-in mouse

*Mfn2$^{R707W}$* mice were generated using CRISPR-Cas9 microinjection of fertilised oocytes at The Wellcome Trust Centre for Human Genetics (Oxford, UK). Two single guide RNA (sgRNA) sequences targetting exon 18 of the mouse *Mfn2* gene (ENSMUSE00000184630) were used, namely 5'- CACC gTTCCTGCTCCAGATTATCTC-3' and 5'-AAACGAGATAATCTGGAGCAGGAAc-3'. The single-strand donor oligonucleotide (ssODN) incorporated the desired R707W mutation by recoding codon 707 from CGA to TGG. (This involved two point mutations in order to avoid a stop codon.) A silent mutation was added upstream to generate an EcoRV restriction site (*Figure 1*).

Superovulated 3-week-old C57BL/6 J female mice were mated with C57BL/6 J males. Embryos were extracted on day 0.5 of pregnancy and cultivated until two pronuclei were visible. One pronucleus was injected with purified sgRNA (20 ng/µL), Cas9 protein (100 ng/µL), and the ssODN template (10 ng/µL). Embryos were reimplanted into pseudopregnant CD1 foster mothers at day 0.5 post-coitum. Mfn2$^{R707W}$ was confirmed by Sanger sequencing of F0 founder males (with one additional upstream silent mutation ACC >ACA). Following cryopreservation of embryos, the line was re-derived in a colony of C57BL/6 J mice in Cambridge, UK.

Genotyping utilised the upstream EcoRV restriction enzyme digestion site. For genotyping, ear biopsies were digested in Chelix and proteinase K (0.1 mg/mL) for 45 min at 50 °C to extract gDNA for use in PCR. gDNA was amplified using primers 5'-AGTCCCTTCCTTGTCACTTAGT-3' and 5'-ATCTCACAAGAAAGCGAAATCC-3' and GoTaq DNA Polymerase (Promega), then digested using EcoRV (New England Biosciences). Wild-type (WT) *Mfn2* generates a PCR product of 523 bp, homozygous knock-ins (KI) have two bands at 327 bp and 196 bp, and heterozygotes have all three bands (*Figure 1*).

## Mouse husbandry and phenotyping

All experiments were performed under UK Home Office-approved Project License 70/8955 except for thermogenic capacity assessments which were conducted under P0101ED1D. Protocols were approved by the University of Cambridge Animal Welfare and Ethical Review Board. Animals were co-housed in groups of 2–5 littermates of mixed genotype, on 12 hr light/dark cycles. They had access to food and water ad libitum except when fasting prior to experimental procedures.

Separate cohorts of male and female mice were studied. Unless otherwise stated (*Figure 3—figure supplement 2*), data represents results from male mice. WT and KI male mice aged 5 weeks were randomly allocated to HFD (45% kcal as fat, 4.7kcal/g, Research Diets D12451i) or chow (Safe Diets R105-25) for up to 6 months. Investigators were blinded to animal genotype at the point of data collection. Mice were weighed weekly. Tail blood samples were collected four weekly into heparinised capillary tubes (Hawksley) and spun at 13,000 g for 4 min for plasma analysis of leptin and adiponectin. Six hr fasted blood samples were obtained on weeks 16 and 24 of diet, and prior to sacrifice, for analysis of glucose, insulin, and lactate, whilst other blood samples were from fed animals. For lactate, tail vein blood was collected into fluoride oxalate tubes and centrifuged immediately before freezing of plasma at –80 °C.

An intraperitoneal glucose tolerance test (IPGTT) was performed on week 31 of chow and HFD and an intraperitoneal insulin tolerance test (IPITT) was performed on week 32 of HFD (only) after a 6 hr fast. One g/kg of glucose and 0.75 units/kg of insulin were administered for the IPGTT and IPITT, respectively. Blood glucose was measured at 0, 10, 20, 30, 60, 90, and 120 min after the injections using a glucometer (Abbot Laboratories) and glucose test strips (Abbot Laboratories). Insulin was measured at 0 min at the start of IPGTT from a tail vein blood sample.

Body composition (lean and fat mass) was assessed prior to sacrifice by Time-Domain Nuclear Magnetic Resonance (TD-NMR) using a Minispec Live Mouse Analyzer (Bruker). Mice were sacrificed at 4- or 33 weeks on diet after a 6 hr fast. Tissues were weighed, sections were removed for histological or electron microscopic analysis, and remaining tissue was snap frozen in liquid nitrogen.

To estimate the number of mice required for experimental groups, We used data from the *Adipoq*::Cre *Mfn2* knock-out mouse (*Boutant et al., 2017*), aiming to determine a difference between fat mass in Mfn2[R707W] and wild-type. Mean fat mass in adipose-specific *Mfn2* knock-out=3.8 ± 0.43 g. Mean fat mass in wild-type=2.9 ± 0.14 g. For 80% power at 0.05 significance, to detect 0.9 g difference, sample size: 8 animals per group.

## Calorimetry studies

Eight-week-old chow-fed male mice were housed in either cold (10 °C) or thermoneutrality (30 °C) for 4 weeks. Animals were anaesthetised using 90 mg/kg of pentobarbital by intraperitoneal injection and were placed in 2.7 l calorimetry chambers (Oxymax, Columbus instruments, Ohio) attached to a Promethion calorimetry system (Sable Systems, Las Vegas, NV, USA) pre-warmed to 30 °C for 20 min for measurement of basal energy expenditure. They were then given a subcutaneous injection of 2 µl/g of 0.5 mg/mL noradrenaline bitartrate (NA) plus 1.66 µl/g of 18 µg/µl pentobarbital and NA-stimulated energy expenditure was measured for 25 min. Animals were then sacrificed, and tissues snap frozen as described above. Basal energy expenditure was calculated from the three readings prior to NA-injection. Peak energy expenditure was calculated from the three greatest readings 5–25 min after NA-injection. NA-induced energy expenditure was calculated as the difference between peak and basal energy expenditure.

## Transmission electron microscopy (TEM)

Chow-fed mice aged 8 weeks were sacrificed and white adipose tissue (inguinal and epididymal), brown adipose tissue, skeletal muscle (quadriceps), heart, and liver were removed, cut into <1 mm³

cubes and fixed (2% glutaraldehyde/2% formaldehyde in 0.05 M sodium cacodylate buffer pH 7.4 containing 2 mM calcium chloride) on a rocker at 4 °C overnight. Samples were then washed five times with 0.05 M sodium cacodylate buffer pH 7.4 and osmicated (1% osmium tetroxide, 1.5% potassium ferricyanide, 0.05 M sodium cacodylate buffer pH 7.4) for 3 days at 4°C.

Following initial osmication, samples were washed five times in DIW (deionised water) then treated with 0.1% (w/v) thiocarbohydrazide/DIW for 20 min at room temperature in the dark. After washing five times in DIW, samples were osmicated a second time for 1 hr at room temperature (2% osmium tetroxide/DIW). After washing five times in DIW, samples were block stained with uranyl acetate (2% uranyl acetate in 0.05 M maleate buffer pH 5.5) for 3 days at 4 °C. Samples were washed five times in DIW and then dehydrated in a graded series of ethanol (50%/70%/95%/100%/100% dry) 100% dry acetone and 100% dry acetonitrile, three times in each for at least 5 min. Samples were infiltrated with a 50/50 mixture of 100% dry acetonitrile/Quetol resin (without benzyldimethylamine (BDMA)) overnight, followed by 3 days in 100% Quetol (without BDMA). Then, samples were infiltrated for 5 days in 100% Quetol resin with BDMA, exchanging the resin each day. The Quetol resin mixture is: 12 g Quetol 651, 15.7 g NSA, 5.7 g MNA and 0.5 g BDMA (all from TAAB). Samples were placed in embedding moulds and cured at 60 °C for 3 days.

Thin sections were cut using an ultramicrotome (Leica Ultracut E) and placed on bare 300 mesh copper TEM grids. Samples were imaged in a Tecnai G2 TEM (FEI/Thermo Fisher Scientific) run at 200 keV using a 20 µm objective aperture to improve contrast. Images were acquired using an ORCA HR high-resolution CCD camera (Advanced Microscopy Techniques Corp, Danvers USA).

Analysis was performed by manual measurement of individual mitochondrion from all obtained images using Fiji (*Schindelin et al., 2012*) /ImageJ measurement tools by an investigator who was blinded to the genotype of tissues/cells. In all tissues, mitochondrial perimeter and aspect ratio (length/ width) were determined. In addition, in brown adipose tissue, the number of cristae per mitochondrion and mitochondrial-lipid droplet contacts were quantified. There were insufficient mitochondrial-lipid droplet contact sites with high quality preservation to permit this in white adipose tissue. In liver, mitochondrial-endoplasmic reticulum contact sites were quantified. It was not possible to assess for mitochondria-ER contacts in other tissues due to quality of preservation.

## Isolation of mitochondria for Oroboros analysis

For ex vivo measurement of mitochondrial respiratory capacity, mitochondria were isolated from BAT and liver of 12-week-old chow-fed male mice using the protocols from *McLaughlin et al., 2020* for BAT and *Fernández-Vizarra et al., 2010* for liver. In brief, tissues were isolated and washed in buffer 'B' then homogenised using a drill-driven Teflon pestle and borosilicate glass vessel. Homogenates were centrifuged at 800 x $g$ for 10 min at 4 °C to remove cellular debris. Supernatant was removed, and then re-centrifuged at 10,000 x $g$ for 10 min at 4 °C to enrich mitochondria. Supernatant was discarded and mitochondria were resuspended in buffer 'A'. Mitochondria were quantified using a BioRad Protein Assay.

Fifty µg of protein were used per chamber for high-resolution respirometry, using Oroboros (Innsbruck, Austria). There was sequential injection of: 20 µl of 1 M glutamate and 10 µl of 1 M malate (liver only), 20 µl of 0.5 M ADP, 20 µl of 1 M pyruvate, 20 µl of 2 M succinate, 1 µl of 1 mM CCCP (liver only), 2 µl of 2 mM rotenone, and 2 µl of 2 mM antimycin A.

## mtDNA content assay

Relative mtDNA content was assayed using real-time quantitative polymerase chain reaction (RT-qPCR) quantification of mitochondrial *Rnr2* and nuclear *Hk2* DNA. DNA was extracted from snap frozen murine tissue, using a DNeasy kit (Qiagen) as per the manufacturer's instructions. DNA was quantified using a Nanodrop and diluted to 4 ng/µl. RT-qPCR was performed in triplicate for each sample using 8 ng DNA with primers for *Hk2* and *Rnr2*. mt*Rnr2*/n*Hk2* was calculated using the standard curve method and expressed relative to WT.

## Protein extraction for WB

Fifty mg of frozen tissue was crushed using a pestle and mortar in liquid nitrogen. Powdered tissue was dissolved in 800 µl of RIPA buffer (Sigma, R0278) containing protease (Sigma, 11836170001) and phosphatase inhibitors (Roche, 04906837001). Samples were sonicated twice for 5 s at 30 Hz before

centrifugation at 10,000 x g for 5 min at 4 °C. Supernatant was extracted and, for adipose samples, re-centrifuged to remove excess lipid.

Thirty-45 µg of protein lysates were mixed with NuPAGE 4 x LDS buffer (ThermoFisher Scientific), containing 0.05% 2-mercaptoethanol, and denatured for 5 min at 95 °C. Samples were run on 4–12% Bis-Tris gels (Invitrogen) and transferred onto a nitrocellulose membrane using iBlot-2 (ThermoFisher Scientific). For Opa1 immunobloting, protein lysates were prepared using 4 x BOLT LDS sample buffer and resolved using BOLT 8% Bis-Tris with BOLT MOPS SDS running buffer (ThermoFisher Scientific). Membranes were washed in Tris-buffered saline with 0.1% (vol/vol) Tween 20 (TBST, Sigma) before blocking in 5% (wt/vol) skimmed milk powder dissolved in TBST. Membranes were incubated with primary antibodies (*Supplementary file 1*) at 4 °C for 16 hr, washed with TBST five times for 5 min, followed by incubation with horseradish peroxidase (HRP)-conjugated secondary antibodies for 1 hr at room temperature. Blots were developed using Immobilon Western Chemiluminescent HRP Substrate (Millipore) with images acquired on BioRad ChemiDoc Imaging system or ImageQuant LAS 4000 (GE Healthcare).

For Opa1 quantification, Oma1-cleaved S-Opa1 bands were calculated as described by *Shammas et al., 2022*, with the intensity of each of the five Opa1 bands on the blot measured., The intensity of c and e bands were summed and divided by the sum of the five bands (a–e) to obtain the percentage of S-Opa1 generated by Oma1 from total Opa1.

## RNA isolation and qPCR analysis

At the end of the study, tissues were harvested and immediately snap frozen in liquid nitrogen and stored at –80 °C. For RNA isolation, 30–50 mg of tissue was placed in Lysing Matrix D tubes and homogenized in 800 µl TRI Reagent (T9424, Sigma) using the Fastprep-24 Homogenizer for 30 s at 4–6 m/s (MP Biomedical). Homogenate was transferred to an RNase free tube and 200 µl chloroform (Sigma) added. The samples were vortexed and centrifuged at 13,000 rpm for 15 min at 4 °C. The upper phase was then transferred to an RNase free tube and mixed with an equal volume of 70% ethanol before loading onto RNA isolation spin columns. RNA was extracted using a RNeasy Mini Kit (74106, Qiagen) isolation kit following the manufacturer's instructions.

Total RNA of 600 ng was quantified using Nanodrop and converted to cDNA using MMLV Reverse Transcriptase with random primers and RNase inhibitor (Promega). RT-qPCR was performed using SYBR Green or TaqMan Universal PCR MasterMixes (Applied Biosystems) on QuantStudio 7 Flex Real time PCR system (Applied Biosystems). Primers are listed in *Supplementary file 2*. Reactions were performed in triplicate and RNA expression was normalised to *36b4*, *Hprt*, and *B2m* expression using the standard curve method.

## Immunoassays

Mouse sera and plasma were analysed by the Cambridge Biochemical Assay Laboratory, University of Cambridge. Leptin was measured using a 2-plex Mouse Metabolic immunoassay kit from Meso Scale Discovery Kit (Rockville, MD, USA) according to the manufacturers' instructions and with supplied calibrants. GDF15 was measured using a modified Mouse GDF15 DuoSet ELISA (R&D Systems) as an electrochemiluminescence assay on the Meso Scale Discovery platform. Adiponectin (K152BYC-2, MSD) was analysed individually using the Meso Scale Discovery Kit (Rockville, MD, USA). NEFA were analysed using the Free Fatty Acid Kit (half-micro test) (11383175001, Roche) and TG was measured using an enzymatic assay (DF69A, Siemens Healthcare). Alanine aminotransferase (ALT, product code DF143), aspartate aminotransferase (AST, product code DF41A), and total cholesterol (product code DF27) were measured using automated enzymatic assays on the Siemens Dimension EXL analyzer. Mouse Fgf21 was measured using an enzyme-linked immunosorbent assay kit (R&D/ Biochne, cat no. MF2100).

## Histological processing

Fresh tissue was fixed in 10% formalin for 24 hr at room temperature immediately following sacrifice. Tissue was then embedded in paraffin and 4 µm sections were cut then baked overnight at 50 °C. For haematoxylin & eosin (H&E) staining, slides were dewaxed in xylene for 5 min twice, then dehydrated in 100% ethanol for 2 min twice. Following a 3-min water wash, slides were stained with filtered Mayer's haematoxylin (Pioneer Research Chemicals) for 7 min and blued in water for 4 min. Slides

were then stained with 1% aqueous eosin (Pioneer Research Chemicals) for 4 min and briefly washed in water before dehydrating in 100% ethanol (1 min, twice) and cleared in xylene (2 min, twice) and mounting with Pertex. Slides were imaged using a Axio Scan Z1 slide scanner (Zeiss). Lipid droplet area and hepatic steatosis was quantified automatically using Halo software (Indica Labs).

## Transcriptomic profiling in white and brown adipose tissue

RNA was isolated from three tissues/dietary conditions for RNA sequencing: (1) inguinal WAT from chow fed animals (n=7 WT, n=6 KI, single technical replicates); (2) inguinal WAT from HFD-fed animals (n=8 WT, n=8 KI, two technical replicates per animal); and (3) BAT from HFD-fed animals (n=6 WT, n=6 KI, single technical replicate). RNA was quantified using Agilent 2100 Bioanalyzer (Agilent Technologies Inc) and only samples with RNA Integrity Number ≥8 were used for library preparation. cDNA libraries were made using Illumina TruSeq RNA sample kits and sequencing was performed on Illumina NovaSeq 6000 with paired-end 150 bp reads (Novogene, Cambridge, UK). Raw reads all passed quality control for $Q_{score}$, error rate distribution, and AT/GC distribution.

Adapter sequences were removed from raw FASTQ files using cutadapt *Martin, 2011* and aligned to *Mus musculus* reference genome (GRCm38) using STAR *Dobin et al., 2013*. Binary alignment/map (BAM) files were sorted using samtools *Danecek et al., 2021* and counts were performed using featureCounts *Liao et al., 2014*. Differential gene expression (DGE) between WT and KI was performed using DESeq2 *Love et al., 2014*, where significance was considered as a Benjamini-Hochberg false-discovery rate (FDR) corrected p-value <0.01. Pathway analysis was performed with the EnrichR package for R *Xie et al., 2021*; *Chen et al., 2013*; *Kuleshov et al., 2016* using significantly differentially expressed genes to determine enriched Hallmark *Liberzon et al., 2015* and Kyoto Encyclopaedia of Genes and Genomes (KEGG) *Kanehisa and Goto, 2000* gene sets. Gene sets with FDR-corrected p-value <0.05 were considered enriched. Figures were generated in R 4.0.2 *R Development Core Team, 2019* using packages pheatmap, ggplot2, and dplyr.

## Adipose explant experiments

Inguinal (subcutaneous) and epididymal (visceral) adipose tissue was harvested from 12-week-old male C57BL/6 J mice fed either chow or HFD for 4 weeks (i.e. 8–12 weeks of age). Tissue was placed in Hanks' Balanced Salt Solution (HBSS, H9269, Sigma) and kept on ice before cutting into 1–2 mm fragments. Approximately 100 mg fragments were incubated in a 12-well plate with M199 media ± 7 nM insulin (Actrapid, Novo Nordisk) and 25 nM dexamethasone (D4902, Sigma). After 24 hr incubation, media were collected, spun down at 5000 x g and stored at –80 °C until leptin and adiponectin assay as above. Explant tissues were weighed and snap frozen for RNA analysis.

## Primary adipocyte experiments

Mature white adipocytes were isolated from 12- to 20-week-old chow-fed male or female C57BL/6 J or C57BL/6 N mice as previously described *Li et al., 2021*; *Harms et al., 2019* with modifications. Briefly, after dissection, gonadal adipose tissue was washed, minced finely, and dissociated to a single-cell suspension in Hanks' Balanced Salt Solution containing 2.25% (w/v) BSA (Sigma, Cat. H9269) and 1 mg/mL collagenase (Sigma, C6885-1G) for approximately 15 min at 37 °C in a shaking incubator. Digested material was diluted with 6 x volume of high glucose DMEM media (Sigma, Cat. D6546) and connective tissue or undigested pieces were removed by passing through a 100 µm nylon mesh (Fisherbrand, 11517532). Floating adipocytes were washed twice with 6 x volume of high glucose DMEM media before being used for downstream experiments.

Mature adipocytes with a packed volume of 60 µL were then cultured in 500 µL of high glucose DMEM media (supplemented with 10% FBS [Gibco Cat. 10270–106], 2 mM L-glutamine [Sigma Cat. G7513] and 1 x Penn/Strep [Sigma Cat. P0781]) per well in the presence of 100 nM insulin in a 24-well plate. On the third day of in vitro culture, 100 µL of media was sampled followed by media replenishment. Cells were treated with indicated concentration of Thapsigargin or Tunicamycin for 6 hr. Media were sampled at the end of treatment and cells collected for subsequent RT-qPCR or western blotting. Adipokine secretion from adipocytes during the 6 hr window was calculated as X=B A*400/500 (A/B=medium adipokine concentration before/after treatment).

## Statistical analysis

Continuous data were expressed as mean ± standard error (SE). Normally distributed data were analysed by t-test (for two group pairwise comparison) and one-way ANOVA (for three or more groups) with post-hoc Bonferroni multiple comparisons test. FDR-corrected p-value <0.05 was considered significant.

Statistical tests of TEM data (pairwise comparisons of mutant to WT) were based on the mean of independent biological replicates (i.e. the number of different samples, not the number of mitochondria) and FDR adjustment for the number of tests (*Lord et al., 2020*).

All experiments were conducted at least three times using, where possible, randomisation of sample order and blinding of experimenters handling samples. No data were excluded from analysis. Data were nalysed using R 4.0.2 (*R Development Core Team, 2019*) and GraphPad Prism version 9 (GraphPad, San Diego). Figures were made using BioRender.

## Materials availability

All reagents used are publicly available. Primer sequences and antibodies are detailed in *Supplementary files 1 and 2*. Code used in analysis is available from: https://doi.org/10.5281/zenodo.5770057. Raw counts from transcriptomic analysis are available from GEO under accession ID: GSE210771.

## Acknowledgements

We are grateful to the team at the University of Cambridge Advanced Imaging Centre for access to their facilities and for their expertise. We thank the Disease Model Core from the Wellcome-MRC Institute of Metabolic Science for their technical assistance in animal work. We also thank the Genomics and Transcriptomics core, the Histology core and G Strachan from the Imaging core for technical assistance. Finally we thank M Mimmack for his assistance with in vitro experiments. This work was supported by the Wellcome Trust (grant Nos. WT 210752 to RKS, WT 219417 to DBS, WT 216329/Z/19/Z to JPM, and WT 214274 to S'R, respectively), the National Institute for Health Research (NIHR) Cambridge Biomedical Research Centre, the Swedish Research Council (to IL) and the Medical Research Council UK (MRC; MC_UU_00015/7 and MC_UU_00028/5 to JP). The Disease Model Core, Biochemistry Assay Lab, the Histology Core and the Genomics and Transcriptomics Core are funded by MRC MC_UU_00014/5, MRC MRC_MC_UU_12012/5, and a Wellcome Trust Strategic Award (208363/Z/17/Z). LCT was a recipient of a Ramon Areces postdoctoral fellowship. The funders had no role in study design, data collection and interpretation, or submitting the work for publication.

## Additional information

### Funding

| Funder | Grant reference number | Author |
|---|---|---|
| Wellcome Trust | 210752 | Robert K Semple |
| Wellcome Trust | 219417 | David B Savage |
| Wellcome Trust | 216329/Z/19/Z | Jake P Mann |
| Wellcome Trust | 214274 | Stephen O'Rahilly |
| Swedish Research Council | | Ineke Luijten |
| Medical Research Council | MC_UU_00015/7 and MC_UU_00028/5 | Julien Prudent |
| Medical Research Council | MC_UU_00014/5 | Stephen O'Rahilly |
| Medical Research Council | MRC_MC_UU_12012/5 | Stephen O'Rahilly |
| Wellcome Trust | 208363/Z/17/Z | Stephen O'Rahilly |
| Ramón Areces Foundation | | Luis Carlos Tábara |

| Funder | Grant reference number | Author |
|---|---|---|

The funders had no role in study design, data collection and interpretation, or the decision to submit the work for publication. For the purpose of Open Access, the authors have applied a CC BY public copyright license to any Author Accepted Manuscript version arising from this submission.

## Author contributions

Jake P Mann, Data curation, Formal analysis, Funding acquisition, Investigation, Methodology, Writing – original draft, Project administration, Writing – review and editing; Xiaowen Duan, Satish Patel, Fabio Scurria, Anna Alvarez-Guaita, Afreen Haider, Data curation, Formal analysis, Investigation, Methodology, Writing – review and editing; Luis Carlos Tábara, Ineke Luijten, Data curation, Formal analysis, Funding acquisition, Investigation, Methodology, Writing – review and editing; Matthew Page, Martin Armstrong, Supervision, Investigation, Methodology, Writing – review and editing; Margherita Protasoni, Investigation, Methodology, Writing – review and editing; Koini Lim, Formal analysis, Investigation, Writing – review and editing; Sam Virtue, Conceptualization, Supervision, Funding acquisition, Methodology, Project administration, Writing – review and editing; Stephen O'Rahilly, Julien Prudent, Conceptualization, Supervision, Funding acquisition, Investigation, Methodology, Project administration, Writing – review and editing; Robert K Semple, David B Savage, Conceptualization, Supervision, Funding acquisition, Investigation, Methodology, Writing – original draft, Project administration, Writing – review and editing

## Author ORCIDs

Jake P Mann ![ORCID] http://orcid.org/0000-0002-4711-9215
Xiaowen Duan ![ORCID] http://orcid.org/0000-0003-1415-8899
Satish Patel ![ORCID] http://orcid.org/0000-0002-5345-8942
Sam Virtue ![ORCID] http://orcid.org/0000-0002-9545-5432
Stephen O'Rahilly ![ORCID] http://orcid.org/0000-0003-2199-4449
Julien Prudent ![ORCID] http://orcid.org/0000-0003-3821-6088
Robert K Semple ![ORCID] http://orcid.org/0000-0001-6539-3069
David B Savage ![ORCID] http://orcid.org/0000-0002-7857-7032

## Ethics

All experiments were performed under UK Home Office-approved Project License 70/8955 except for thermogenic capacity assessments which were conducted under P0101ED1D. Protocols were approved by the University of Cambridge Animal Welfare and Ethical Review Board.

## Decision letter and Author response

Decision letter https://doi.org/10.7554/eLife.82283.sa1
Author response https://doi.org/10.7554/eLife.82283.sa2

# Additional files

## Supplementary files

- Supplementary file 1. Antibodies used in this study.
- Supplementary file 2. Primer sequences used in this study. Fwd, forward primer; Rv, reverse primer.
- MDAR checklist

## Data availability

All reagents used are publicly available. Primer sequences and antibodies are detailed in Supplementary files 1 and 2. Code used in analysis is available from: https://doi.org/10.5281/zenodo.5770057. Raw counts from transcriptomic analysis are available from GEO with accession number GSE210771.

The following datasets were generated:

| Author(s) | Year | Dataset title | Dataset URL | Database and Identifier |
|---|---|---|---|---|
| Mann JP, Savage DB, Semple RK | 2022 | RNAseq from Mfn2-R707W knock-in mice | https://www.ncbi.nlm.nih.gov/geo/query/acc.cgi?acc=GSE210771 | NCBI Gene Expression Omnibus, GSE210771 |
| Mann JP | 2023 | PublishedCode_Mann | https://doi.org/10.5281/zenodo.4656979 | Zenodo, 10.5281/zenodo.4656979 |

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
