## [Editor Report]

This work describes a mouse model of a human mitofusin 2- related lipodystrophy, generated by knockin of Mfn2 R707W, and reports adipocyte-specific effects involving activation of a cellular integrated stress response and consequently reduced secretion of leptin and adiponectin. The phenotypic characterization is thorough, and the data are convincing. The work provides important information to link mitochondrial perturbation selectively with altered adipose function, and to show how these effects contribute to metabolic disease.

---

## [Decision Letter]

**Decision letter after peer review:**

Thank you for submitting your article "A mouse model of human mitofusin 2-related lipodystrophy exhibits adipose-specific mitochondrial stress and reduced leptin secretion" for consideration by *eLife*. Your article has been reviewed by 3 peer reviewers, one of whom is a member of our Board of Reviewing Editors, and the evaluation has been overseen by Carlos Isales as the Senior Editor. The reviewers have opted to remain anonymous.

Essential revisions:

1) Please address the comments of Reviewer 2 to strengthen the conclusion that the integrated stress response is activated and/or to distinguish this from simple ER stress.

2) Further analyses to determine if mitochondria-ER associations are reduced in the mutant mice are warranted (Reviewer 2)

3) Additional data from other tissues, such as liver, would be useful to help understand the tissue-specificity of the effect of the R707W mutation (Reviewer 3).

*Reviewer #1 (Recommendations for the authors):*

Since mTorc may be involved, it would be reasonable to treat the KI mice with sirolimus (or rapamycin) and test whether this increases leptin and adiponectin levels, or whether it rescues any other aspect of the phenotype. This would strengthen the hypothesis that upregulation of mTorc1 signaling contributes to the phenotype.

The hypothesis that secreted proteins are selectively affected is intriguing, and if this is the case then it might be that the proteins that place the greatest demand on the secretory pathway are most affected. Certainly, adiponectin falls in this category, since it is very abundant and must form collagen-like trimers and then oligomerize into higher molecular weight forms. Leptin may also be secreted rapidly. Are there alterations in collagen secretion, and could this contribute to the phenotype in humans? In particular, work from the Scherer lab has implicated collagen VI in adipose biology and insulin sensitivity. It is acknowledged that because the mice do not recapitulate the lipodystrophy observed in humans, and because their insulin sensitivity was not altered, it may be that an effect on collagens would not be observed in the KI mice.

Does induction of the ISR affect secretion of TGF-β family members? Can evidence for altered TGF-β signaling be generated?

Only male mice were used for the study. Can any data from females be included to confirm that a similar phenotype is present, at least for one or two specific aspects of the phenotype?

*Reviewer #2 (Recommendations for the authors):*

The suggestions to address the weaknesses are as follows:

1) To demonstrate that the changes in mitochondria induced by the R707W mutation activate the ISR and that the ISR is responsible to decrease leptin and adiponectin expression, authors should measure:

(1.a.) OMA1 activity: two recent papers demonstrate that OMA1 activation is the mitochondrial factor that senses mitochondrial stress and initiates ISR activation via HRI kinase (Fessler et al. 2020; Guo et al., 2020). OMA1 activation can be easily measured by determining OPA1 processing by Western blot of tissues. That would support specific mitochondrial ISR engagement.

(1.b.) Measure adiponectin protein content in BAT and WAT, instead of measuring mRNA levels. The phosphorylation of eif-2alpha blocks the translation of multiple proteins and if ISR is engaged, it would be expected that decreased adiponectin and leptin are explained by a decrease in translation, rather than ATF4 blocking their transcription.

Final confirmation would be to delete OMA1 or ATF4, but the reviewer understands that this is not possible as it would involve generating new mouse strains. This limitation should be emphasized when discussing the role of ISR controlling adiponectin content in the mutants.

2) Based on ample evidence previously showing that Mfn2 deletion causes ER stress (ref 35,63), it is likely that simple ER stress, rather than the ISR, is explaining the specific effects on adiponectin and leptin secretion caused by this Mfn2 mutation. Accordingly, the transcriptomics analyses pick up unfolded protein response as a major upregulated pathway, rather than ISR, HRI or an ATF4 specific signature. Treating mice with TUDCA to alleviate ER stress systemically as performed in reference 63 and determine whether TUDCA treatment can restore leptin levels in vivo. Another option would be to treat tissue explants from the Mfn2 mutants with TUDCA and measure adiponectin/leptin content and secretion.

3) Mfn2 was shown to tether mitochondria with the ER (de Brito et al., Nature). Given that the authors fail to reproduce that Mfn2 interacts with PLIN1 observed in reference 35, it would be worth analyzing the EM to determine whether mitochondria-ER interactions are decreased in these Mfn2 mutants.

*Reviewer #3 (Recommendations for the authors):*

1) Many of the results hint toward the possibility that the Mfn2R707W generates a partial impairment in Mfn2 function. This, however, is a very vague concept and should be well characterized, through evaluation of Mfn2 GTPase activity and through work in cultured cell models where mitochondrial dynamics, position and interactions can be better traced.

2) The above point is quite important, as Mfn2 deficiency in the liver has been shown to dramatically exacerbate diet-induced metabolic damage. Mfn2 deficiency in POMC led to higher food intake. These phenotypes, however, are not seen here. In contrast, the R707W mutation to some extent recapitulates the effects of the adipose-tissue specific deletion of Mfn2 reported by the Shirihai and Canto labs. The causes behind tissue-specificity in the impact of the R707W mutation should be further investigated.

3) The brown adipose tissue of the knockin mice also seems to display a decrease in Mfn1, which complicates the interpretation of the phenotype. However, there is no information on the age at which these experiments were performed. There could be a possibility that this decrease is secondary to the alterations caused by the knockin mutation. Could the authors evaluate if at a younger age (e.g.: 4 weeks of age) the decrease in Mfn1 is already present? If not, it could be a more interesting point to evaluate the brown adipose tissue phenotype.

4) Cristae density seems affected in multiple tissues and should be quantified. Similarly, OPA-1 processing should also be analyzed through western blot.

5) The authors observe a severe decrease in Complex I, III and IV proteins, yet no decrease in mitochondrial complex I related respiration or in maximal respiratory capacity. How is this even possible?

6) The fact that, according to the authors, there is not mitochondrial respiratory capacity impairment, yet a slight thermogenic impairment is seen, suggests that these mice have impaired ability to mobilize fatty acids for oxidation upon cold-exposure. Did the authors evaluate if there are alterations in adrenergic signaling?

7) The authors make a strong point on the reductions in circulating leptin and adiponectin. However, do these mice show altered food intake, as should be predicted from a large decrease in leptin?

8) Given the focus on leptin and adiponectin, the authors should try to investigate if they are the root of the complications seen in the Mfn2 knockin mice. This could be done by injecting animals with leptin and evaluating if they recover OXPHOS protein levels.

---

## [Author Response]

Essential revisions:1) Please address the comments of Reviewer 2 to strengthen the conclusion that the integrated stress response is activated and/or to distinguish this from simple ER stress.

We think that this issue is partly semantic, relating to the definition of the term integrated stress response (ISR). In our view, activation of the ISR is essentially defined by EIF2a phosphorylation – we demonstrated that this was strongly present in BAT and WAT in Figure 4, so maintain that we have evidence for activation of the ISR in tissues from the R707W KI mice. We suspect that the reviewer is primarily concerned with the mechanism or pathway by which the ISR was activated – ie was it primarily activated by mitochondrial dysfunction or instead by a UPR-related mechanism. Our data did not directly address this question and so we agree that we are not in a position to specify the origin of activation of the ISR.

In order to try to determine exactly what triggered the adipose ISR we describe, we have devoted considerable further effort to western blotting for Opa1, as suggested by the reviewer. This is in fact more complicated than the reviewer implies, particularly in adipose tissue. Indeed we could not find any papers in which all five Opa1 bands are clearly shown in adipose lysates (white or brown) – papers typically only show two or three bands, which does not provide adequate information to inform on Opa1’s activation state (eg PMID 33944779). In other tissues like heart or skeletal muscle five bands are more readily detected and one can then infer Opa1 activity using the approach reported by Shammas et al. (PMID 35700042). We contacted that group and used their protocols in an attempt to optimise the blots. These data are now included in the revised manuscript (Figure 4-Supplementary Figures 3-4). The data do not show any differences in most tissues.

In an attempt to validate the Opa1 blot data, we also undertook Oma1 immunoblotting, as Oma1 typically cleaves itself when activated, so one might expect its expression to fall. These data are also included (Figure 4-Supplementary Figures 3-4) and suggest that, at least in BAT, Oma1 may have been activated, albeit weakly.

The WAT data are more difficult to interpret definitively as the Opa1 blots do not offer sufficient resolution to detect a subtle difference. We also do not see any evidence for a change in Oma1 expression.

So what can we conclude about the origin(s) of ISR activation in this model? In our view, it remains unclear. We do not think that one can definitively exclude a mitochondrial origin for the activation as whilst we agree that the two papers the reviewer mentions do provide compelling evidence for an Oma1-triggered ISR activation pathway, we and others think that there might be additional ways that mitochondrial perturbation could trigger the ISR.

We have no objection to the idea that a UPR-mediated pathway is involved in our mice instead or as well, but we do not have direct evidence for this and have not seen definitive evidence from the Mfn2 KO models to clearly distinguish a UPR from a mitochondrial triggered ISR response.

We have revised the relevant section of the manuscript to reflect all these points. In short, we are convinced that the ISR is selectively activated in BAT and WAT in the KI mice but exactly how it was activated remains unclear.

In relation to the suggestion to check leptin and adiponectin expression in tissue lysates – thank-you for this helpful suggestion. We have done this and included the data which show that both leptin and adiponectin protein expression are reduced in WAT (Figure 5 —figure supplement 2).

2) Further analyses to determine if mitochondria-ER associations are reduced in the mutant mice are warranted (Reviewer 2)

We did report on mitochondrial-ER contacts in liver samples in the original manuscript – there were no differences. Similar analysis is much more challenging in adipose tissue unfortunately. We attempted it on the original sections available as well as on sections of freshly isolated samples but the ER mitochondrial contacts were again inadequately resolved to permit this analysis. Again we are not aware of other papers having reported on this in adipose tissue, at least in Mfn2-related models. We insert representative images in Author response image 1.

**Author response image 1. sa2fig1:** Representative images from transmission electron microscopy on liver and brown adipose tissue from wild-type and Mfn2-R707W knock-in mice. Endoplasmic reticulum (ER) could be clearly identified and mitochondrial-ER contacts quantified in liver (top), with no difference by genotype. In white and brown adipose tissue there was poor preservation of ER with very infrequent mitochondrial-ER contacts, such that a formal assessment was not possible. Data representative from n=4 (liver) and n=8 (adipose tissue) mice.

3) Additional data from other tissues, such as liver, would be useful to help understand the tissue-specificity of the effect of the R707W mutation (Reviewer 3).

We have now added additional data specifically from the liver. In general, we do not see changes in mitochondrial morphology, evidence for ISR activation or changes in gene expression in the liver in the KI mice. We have now included the new gene expression data in the manuscript as requested (Figure 3—figure supplement 1I).

Reviewer #1 (Recommendations for the authors):Since mTorc may be involved, it would be reasonable to treat the KI mice with sirolimus (or rapamycin) and test whether this increases leptin and adiponectin levels, or whether it rescues any other aspect of the phenotype. This would strengthen the hypothesis that upregulation of mTorc1 signaling contributes to the phenotype.

Thank-you, we agree that this is worth considering but feel that it is beyond the scope of this manuscript.

The hypothesis that secreted proteins are selectively affected is intriguing, and if this is the case then it might be that the proteins that place the greatest demand on the secretory pathway are most affected. Certainly, adiponectin falls in this category, since it is very abundant and must form collagen-like trimers and then oligomerize into higher molecular weight forms. Leptin may also be secreted rapidly. Are there alterations in collagen secretion, and could this contribute to the phenotype in humans? In particular, work from the Scherer lab has implicated collagen VI in adipose biology and insulin sensitivity. It is acknowledged that because the mice do not recapitulate the lipodystrophy observed in humans, and because their insulin sensitivity was not altered, it may be that an effect on collagens would not be observed in the KI mice.

This is an interesting idea but we are not aware of a straightforward way in which to evaluate collagen secretion. If the reviewer can provide specific suggestions, we would be willing to consider these.

Does induction of the ISR affect secretion of TGF-β family members? Can evidence for altered TGF-β signaling be generated?

We agree that this is a very interesting area for future investigation but not one which can be readily and conclusively addressed so we feel that it is beyond the scope of this manuscript.

Only male mice were used for the study. Can any data from females be included to confirm that a similar phenotype is present, at least for one or two specific aspects of the phenotype?

We have now included a more limited set of data from a smaller cohort of female mice. Again there were no differences in body weight, glucose and insulin, whereas adiponectin was clearly reduced and leptin tended to be lower. The blood samples were collected in the morning so there was considerable variability in the leptin data which is we think why the visually apparent difference is not statistically significant These data are now mentioned in the manuscript and included as a supplementary figure (Figure 3—figure supplement 2).

Reviewer #2 (Recommendations for the authors):The suggestions to address the weaknesses are as follows:1) To demonstrate that the changes in mitochondria induced by the R707W mutation activate the ISR and that the ISR is responsible to decrease leptin and adiponectin expression, authors should measure:(1.a.) OMA1 activity: two recent papers demonstrate that OMA1 activation is the mitochondrial factor that senses mitochondrial stress and initiates ISR activation via HRI kinase (Fessler et al. 2020; Guo et al., 2020). OMA1 activation can be easily measured by determining OPA1 processing by Western blot of tissues. That would support specific mitochondrial ISR engagement.

Thank-you, this point was addressed above.

(1.b.) Measure adiponectin protein content in BAT and WAT, instead of measuring mRNA levels. The phosphorylation of eif-2alpha blocks the translation of multiple proteins and if ISR is engaged, it would be expected that decreased adiponectin and leptin are explained by a decrease in translation, rather than ATF4 blocking their transcription.

Thank-you, again this point was addressed above.

Final confirmation would be to delete OMA1 or ATF4, but the reviewer understands that this is not possible as it would involve generating new mouse strains. This limitation should be emphasized when discussing the role of ISR controlling adiponectin content in the mutants.

Thank-you. We have edited the text to reflect uncertainty about the origins of ISR activation in the revised manuscript.

2) Based on ample evidence previously showing that Mfn2 deletion causes ER stress (ref 35,63), it is likely that simple ER stress, rather than the ISR, is explaining the specific effects on adiponectin and leptin secretion caused by this Mfn2 mutation. Accordingly, the transcriptomics analyses pick up unfolded protein response as a major upregulated pathway, rather than ISR, HRI or an ATF4 specific signature.

This general point was discussed in more detail above. However, we also point out that the transcriptomic analysis does not distinguish between ISR and UPR. The Hallmark UPR gene set used in our analysis (http://www.gsea-msigdb.org/gsea/msigdb/human/geneset/HALLMARK_UNFOLDED_PROTEIN_RESPONSE.html) includes both ‘upstream’ UPR genes (e.g. *ATF6*, *EIF2AK3* (encoding PERK), *XBP1*, *ERN1* (encoding IRE1), ‘downstream’ ISR genes (e.g. *ATF3*, *ATF4*), and genes induced by mitochondrial stress (e.g. *MTHFD2*)). Furthermore, activation of the ISR is a major downstream arm of the UPR, so one cannot really draw significant conclusions about the origins of ISR activation from this analysis.

Treating mice with TUDCA to alleviate ER stress systemically as performed in reference 63 and determine whether TUDCA treatment can restore leptin levels in vivo. Another option would be to treat tissue explants from the Mfn2 mutants with TUDCA and measure adiponectin/leptin content and secretion.

Thank-you. This is an interesting suggestion but we feel is beyond the scope of the current manuscript.

3) Mfn2 was shown to tether mitochondria with the ER (de Brito et al., Nature). Given that the authors fail to reproduce that Mfn2 interacts with PLIN1 observed in reference 35, it would be worth analyzing the EM to determine whether mitochondria-ER interactions are decreased in these Mfn2 mutants.

This point was addressed in the Essential revisions section above.

Reviewer #3 (Recommendations for the authors):1) Many of the results hint toward the possibility that the Mfn2R707W generates a partial impairment in Mfn2 function. This, however, is a very vague concept and should be well characterized, through evaluation of Mfn2 GTPase activity and through work in cultured cell models where mitochondrial dynamics, position and interactions can be better traced.

Whilst we agree that it would be very helpful to understand exactly how the R707W mutation perturbs Mfn2 function, we suspect that this will be challenging and require a lot more work, which we feel is beyond the scope of this manuscript. The function of the HR2 domain, in which the mutation sits, remains unclear in general so it is hard to design assays that would specifically test the impact of the mutation. If the reviewers have specific suggestions for us, we would be willing to consider these.

2) The above point is quite important, as Mfn2 deficiency in the liver has been shown to dramatically exacerbate diet-induced metabolic damage. Mfn2 deficiency in POMC led to higher food intake. These phenotypes, however, are not seen here. In contrast, the R707W mutation to some extent recapitulates the effects of the adipose-tissue specific deletion of Mfn2 reported by the Shirihai and Canto labs. The causes behind tissue-specificity in the impact of the R707W mutation should be further investigated.

Whilst we again agree that understanding the tissue specific phenotypes observed is of major interest, it will require a lot more work. The lack of effect on food intake in our KI mice is one observation among several that demonstrates that the R707W variant is not equivalent to a KO. Similarly, the adipose phenotype is distinct from that of adipose-specific KOs, which is interesting but difficult to explain in full at this stage. This condition is further evidence of the value of studying human phenotypes caused by neomorphic or hypomorphic missense alleles which are much less commonly generated in primary mouse studies. Such alleles which are expressed at the protein level often have distinct and sometimes more subtle functional consequences than KO.

3) The brown adipose tissue of the knockin mice also seems to display a decrease in Mfn1, which complicates the interpretation of the phenotype. However, there is no information on the age at which these experiments were performed. There could be a possibility that this decrease is secondary to the alterations caused by the knockin mutation. Could the authors evaluate if at a younger age (e.g.: 4 weeks of age) the decrease in Mfn1 is already present? If not, it could be a more interesting point to evaluate the brown adipose tissue phenotype.

We have done this and included the data which suggest that Mfn1 is also reduced in young mice (See Figure 1—figure supplement 1G). Nevertheless this might still be secondary to the Mfn2 induced perturbation.

4) Cristae density seems affected in multiple tissues and should be quantified. Similarly, OPA-1 processing should also be analyzed through western blot.

Thank-you, Opa1 processing was discussed above. We did not observe any differences in mitochondrial cristae in non-adipose tissue. In white adipose tissue there was insufficient cristae preservation (even in wild-type samples) to allow a formal analysis. However, there is qualitative evidence of altered cristae density in BAT though on statistical analysis the p-value is.07 (see Figure 2E).

5) The authors observe a severe decrease in Complex I, III and IV proteins, yet no decrease in mitochondrial complex I related respiration or in maximal respiratory capacity. How is this even possible?

The reductions in these proteins were in fact modest in BAT (below statistical significance on quantification) which is what we used for the respiratory capacity analyses. WAT has fewer mitochondria so this analysis was not possible in WAT-derived mitochondria. Furthermore we are only showing expression of some components of the complexes so it is possible that expression of others was less significantly perturbed.

6) The fact that, according to the authors, there is not mitochondrial respiratory capacity impairment, yet a slight thermogenic impairment is seen, suggests that these mice have impaired ability to mobilize fatty acids for oxidation upon cold-exposure. Did the authors evaluate if there are alterations in adrenergic signaling?

We studied sizable cohorts of mice in this analysis and the differences were not statistically significant so no we did not proceed to analyse adrenergic signalling in the mice.

7) The authors make a strong point on the reductions in circulating leptin and adiponectin. However, do these mice show altered food intake, as should be predicted from a large decrease in leptin?

In our view the reductions in leptin, whilst clearly statistically significant, do not lead to very low leptin levels. As leptin has its greatest impact on food intake at low levels, we infer that this is why this did not emerge as a striking phenotype in male or female mice.

8) Given the focus on leptin and adiponectin, the authors should try to investigate if they are the root of the complications seen in the Mfn2 knockin mice. This could be done by injecting animals with leptin and evaluating if they recover OXPHOS protein levels.

This is an interesting idea but we think is beyond the scope of this manuscript. Furthermore, as alluded to above, leptin responses are most dramatic when injecting leptin into mice with very low levels or leptin. In our case, leptin was only relatively low in the mice.